# Education for all in action: Measuring teachers' competences for inclusive education

**Wendelien Vantieghem**[1]*, **Iris Roose**[2], **Karin Goosen**[1], **Wouter Schelfhout**[3], **Piet Van Avermaet**[1]

1 Centre for Diversity & Learning, Ghent University, Ghent, Belgium, 2 Human & society, Artevelde University College, Ghent, Belgium, 3 Antwerp School of Education, University of Antwerp, Antwerpen, Belgium

* wendelien.vantieghem@ugent.be

**Data Availability Statement:** The datasets used in the research have been made publicly available via Zenodo. A reference to these datasets is included in the Methodology: Sample & respondents; and the

## Abstract

While inclusive education has achieved international importance, there is no valid instrument to measure teachers' competences in creating quality classrooms for diverse learners, which this study aims to remedy. Exploratory and confirmatory factor analyses with 975 pre-service teachers and 600 in-service teachers were used. Central to teachers' inclusive teaching competency is both beliefs and efficacy. Results show that teachers hold professional beliefs on student diversity, organized in four factors mapping unto axes of diversity (specifically ethnicity, disability, SES, and gender & sexuality). Teachers also hold beliefs on the responsibility of the educational field to create inclusion, organized in three factors: general school policy, initiatives geared specifically towards ethnic minority students, and initiatives for students with a disability. Furthermore, the results show five factors related to self-efficacy: noticing student diversity, enabling high-quality student-interactions, creating stimulating learning environments, collaborating with colleagues and diverse parents. The factorial structure and scale-scores are discussed for what they unveil of teachers' thinking about diversity in the classroom.

## Introduction

Nowadays, due to the expansion of education, migration flows, and various policy initiatives, classrooms are becoming increasingly diverse [1,2]. At the same time, the pursuit of inclusive education is influencing policy agendas of countries across the globe [3,4]. While inclusion was initially focused on groups traditionally occupying a marginalized position in education (including students with disabilities, from ethnic minority descent or low-SES families), inclusive education is currently conceptualised as a call to transform educational systems at large to reach all students regardless of their background [for a detailed discussion of the evolution of the concept of inclusive education, see 5]. This perspective on inclusion explicitly starts from a social justice standpoint and is known as Education for All (EfA) [6]. Important within this view is the recognition of the educational fields' responsibility to take into account differences between learners in order to create qualitative learning environments for everyone, while

DOI's associated with the datasets are DOI: 10.5281/zenodo.8318520 for the dataset with the pre-service teachers and DOI 10.5281/zenodo.8318572 for the dataset with the in-service teachers.

**Funding:** Published with support from the University Foundation (Universitaire stichting) of Belgium to WV.

**Competing interests:** The authors have declared that no competing interests exist.

maintaining a clear understanding of how systemic inequalities in education disproportionally affect specific (groups of) students [7].

EfA can be framed as a so-called wicked problem [8], being a social problem that is both hard to grasp and solve. Wicked problems are characterised by institutional complexity requiring cooperation and changes to several processes, while scientific knowledge on the topic is often fragmented or contested [9,10]. While effective EfA would require changes to curricula, teacher mindsets and behaviour, and cooperation between educational stakeholders (i.e., institutional complexity) [11], most research on EfA employs a narrow focus by considering only one specific diversity group, and is either literature-based or qualitative in nature [12]. While this research has highlighted the crucial role of teachers and has given us some hints on how EfA can be realised, these have not been empirically tested on a large scale due to the lack of valid measurement instruments. Consequently, this article provides insight into the validation of the "DIversity SCreening in educatiOn" (DISCO)-instrument, which is an instrument designed to grasp teachers' competences for inclusive education, and which could help both research and practice move forward in tackling EfA as a wicked problem.

## Teachers' competences in inclusive education

Inclusive teaching has become an obligatory professional competence for teachers in many countries [3]. Unfortunately, teachers both in Flanders and abroad express a low capacity to address the educational needs of diverse pupils [13,14]. Hence, research that provides more insight on teachers' competences for inclusion and the contexts in which these competences can flourish, is sorely needed. Teacher competence for inclusive education is defined as being a complex combination of knowledge, skills and attitudes that allow a teacher to successfully respond to pupils' diversity in the classroom [15]. However, it has to be recognized that developing instruments to grasp teachers' inclusive competences is not an easy task, due to the complex nature of the concept of inclusive education, which entails both a broad scope (i.e., not being focused on one specific group but rather transcending any diversity group), and shows institutional complexity by requiring several educational processes at once to realize maximal learning and developmental outcomes for students [16]. Furthermore, the concept of competence itself has been extensively debated as well. While some researchers have defined competence as actual behaviour in real-world situations, it has been suggested that more can be learned by considering competences as a process which results in observable behaviour while simultaneously acknowledging underlying factors such as knowledge and motivation that underpin and affect this behaviour. Hence, Blömeke, Gustafsson (17) propose to view competences along a continuum with dispositions (such as knowledge and beliefs) and performance being connected through situation-specific skills. These situation-specific skills then entail processes such as perception and decision-making, which determine in a specific situation how dispositions are translated into performance [17,18]. As such, these situation-specific skills are akin to the concept of self-efficacy from social cognitive theory, which proposes that human achievement is dependent on one's behaviours, internal personal factors (such as cognitions and beliefs) and environmental conditions [19,20], and is defined as people's beliefs in their own capability to perform certain tasks [19]. Consequently, in order to get a broad image of aspects central to teachers' competences for inclusive education, research is needed that combines an investigation of teachers' beliefs on diverse learners (i.e., the scope of inclusive education) as well as self-efficacy, which in the context of inclusive education refers to teachers' assessment of their capabilities to create the educational preconditions necessary for students' maximal learning. Both aspects, being beliefs on diversity and sense of self-efficacy, will be discussed in detail below.

## Teachers' beliefs

With regard to teachers' beliefs on diverse learners, two distinct but related aspects are central, namely teachers' professional beliefs on student diversity and teachers' beliefs about the responsibility of education towards diverse learners. With regard to the latter, teachers' view on whether equity is part of the central aims of education and hence, what constitutes the responsibilities of teachers and schools, has a profound impact on shaping their classroom behaviour [21]. However, a meta-review of research in-between 1980 and 2015 shows that teachers tend to report more negative attitudes towards inclusive education than either pupils or parents [4], while a qualitative study shows that teachers who consider it to be the responsibility of education to achieve both excellence and equal learning opportunities, tend to devote themselves more completely to realizing quality classes and supporting struggling students from diverse backgrounds [22]. In short, the degree in which teachers consider providing EfA as one of the educational context's central responsibilities, will be central in shaping their willingness to invest time and effort in the creation of inclusive classrooms. Interestingly, responsibility of teachers cannot only be assessed with regard to learners, but also regarding colleagues and parents [23]. Research suggests that schools with strong parent-teacher connections are better at realizing maximal developmental chances for their students [11], and that achieving equitable learning environments requires a shared responsibility and collaboration among teachers [24]. Consequently, in order to really grasp teachers' competences in creating inclusive education, collaboration with these partners should not be disregarded.

Secondly, teachers' beliefs on learners from diverse backgrounds is of central importance as well, called teachers' professional beliefs about diversity [25]. These beliefs subconsciously influence teacher behaviour, for instance by involving students for whom teacher have lower expectations less actively in classroom activities or by seeing poor performance as normal [26,27]. This process has a negative impact on both students' cognitive outcomes and well-being [28]. Importantly, such effects of teacher expectations tend to be especially powerful for at-risk students, such as students from lower socioeconomic backgrounds and ethnic minority descent [26]. Hence, in the framework of inclusive education, it is particularly important to understand and assess teachers' beliefs on teaching diverse learners, to counter the negative self-fulfilling prophecies for these students. Unfortunately, research on teachers' beliefs towards diverse learners often employs a narrow definition of diversity, for instance by focusing only on teachers' multicultural beliefs [e.g., 29], or beliefs on students with behavioural difficulties [e.g., 30]. While such specific beliefs allow for a fine-grained analysis of educational barriers for particular student groups, it remains unclear how teachers' beliefs towards several groups of students relate to each other and whether an overall inclusive mindset among teachers exists. Consequently, to fundamentally advance the research on classrooms that are effective for all students, research must consider teacher beliefs towards several groups of students simultaneously.

Hence, we propose to assess teachers' beliefs on diverse learners by assessing 1) teachers' professional beliefs towards students along important axes of diversity (i.e., ethnicity, disability, gender & sexual orientation, and socio-economic status) and 2) teachers' beliefs on educational responsibility with regards to diverse students, parents and colleagues.

## Teachers' sense of efficacy

In order to realise the maximal learning and developmental outcomes suggested within EfA, inclusion cannot be relegated to the side-lines as being something that some teachers sometimes do within the context of specific courses or lessons, or something that only expert teachers do after having achieved mastery in "general" teaching tasks aimed at "regular" students

[31]. Rather, it becomes an inclusive culture that all teachers adhere to in the classroom [6], and which requires a constant effort of teachers to continuously monitor, evaluate and adapt their teaching methods to best suit the needs of the students present [32]. In such an endeavour, self-efficacy could constitute a powerful resource, as self-efficacy beliefs have proven to be powerful predictors of teachers' motivation and actual performance [33], with a high sense of self-efficacy being especially powerful in the face of adversity in order to persevere when faced with difficulties or failure [19].

Importantly, research has shown that self-efficacy is not unidimensional, but tends to be context- and task-specific [34]. Consequently, it is important to consider teachers' self-efficacy with regards to several necessary tasks to create inclusive classroom environments rather than one overall assessment. At the same time, it remains important to employ a broad definition of diversity in order to truly capture EfA and the intersectionally diverse nature of classroom contexts nowadays, rather than narrowing the focus to a specific diversity group (note that such teaching self-efficacy scales do exist, such as the ones focused on ethnic diversity developed by Siwatu [35] or Romijn, Slot [36], or focused on students with a disability by Sharma, Loreman [37]). Hence, in this research, teachers' self-efficacy beliefs with respect to several central aspects of realizing inclusive learning environments will be assessed. That is: 1) noticing diversity, 2) creating powerful learning environments for diverse students, and 3) realizing positive classroom interactions for diverse students. The first subcomponent of noticing diversity, taps the extent to which teachers see themselves capable of actively gathering information on several aspects of student diversity (f.i., students' background, interests, well-being, . . .). Only when teachers are aware of and exploit these differences, can a classroom environment be maximally adapted to the strengths and needs of every student [38]. In tandem with this noticing, the second subcomponent taps the extent to which teachers feel capable to design a classroom environment that takes into account the strengths and needs of their students. The third subcomponent measures the extent to which teachers feel capable to create positive classroom climates for their students, since research has shown both the impact teachers have on the creation of qualitative relationships among students [39] as well as the vital importance of this for at-risk students' well-being and achievement [40]. Note that by delineating these aspects, this paper does not explicitly follow the oft-used distinction in teachers' self-efficacy research in-between instructional strategies, classroom organisation and student engagement [41,42]. This choice was fuelled by researchers noting that EfA does not necessarily require new teaching practices, but rather an awareness of diversity and a goal-orientation towards equitable learning opportunities in what teachers do [6,43]. For instance, within the aspect of positive classroom interactions can cooperative learning be linked to both student engagement and instructional strategies [44]. Hence, we felt that an instrument designed to grasp teachers' efficacy for inclusive education would be better served by recognizing this complex intermingling of goals and methods rather than a strict distinction in-between instructional strategies, classroom organisation and student engagement.

As discussed above, there is currently a lack of valid instruments to grasp teachers' competence for inclusive education, due to the institutional complexity and broad scope of EfA. That is, to truly capture competence for EfA, researchers require an instrument that considers both teachers *willingness* to work on inclusion (i.e., beliefs) as well as their perceived *capabilities* to create inclusive learning environment (i.e., self-efficacy). Furthermore, existing instruments tend to focus on either a specific teaching sub-component, such as class management or differentiated instruction, or on a specific student group, such as students with a disability or ethnic minority background, while EfA transcends specific diversity groups and encompasses several educational processes necessary for maximal learning outcomes. Hence, the goal of this article is to establish a valid and reliable instrument to measure teachers' beliefs and self-efficacy with

regards to inclusive education. Furthermore, teachers' scale-scores on this validated instrument are analysed in order to gain insight on teachers' thinking about diversity in the classroom.

## Methodology

### Sample & respondents

In order to allow for a process of trial and improvement as well as cross-validation, data was collected in two steps within Flanders (Belgium). In the first step, data was collected among pre-service teachers. Based on the exploratory factor analyses on this data, the survey was optimised and then tested again in the second step. During this second step, data from a new sample consisting of in-service teachers was analysed using confirmatory factor analyses (for more information, see Plan of analysis). In the first step, data was collected among pre-service teachers in the fall of 2017 in a sample of teacher education institutions [45]. All teacher education institutions within a Flemish province were invited to participate. Seven of the eight institutions decided to participate in the study (participation rate = 87.5%), translating to a total number of 975 pre-service teachers filling out the online research instruments. Of these, 167 students were in training to become a nursery teacher, 155 were studying to become primary school teachers, whereas 143 and 433 were studying to become respectively lower or higher secondary school teachers. 75% of the sample were women, while the mean age was 27.38 with a range from 17 to 57. By the end of the survey, 647 students had provided valid answers, translating to an attrition rate of 33.6%.

In the second step, data was collected with in-service teachers in the spring of 2018 [46]. All schools within one Flemish city were invited to participate. Of the 40 contacted schools, 28 participated in the study, translating to a participation rate of 70%. All teachers within each school were invited to fill out the online survey (N = 600). Of these, 213 were working in a primary school, while 384 were situated in a secondary school (with 3 people providing no answer to this question). 73.2% of the sample were women, while the mean age was 41.02 with a range from 22 to 63. When comparing the data to previous descriptions of the Flemish teaching population [47,48], we see a congruence for both age and gender divisions, indicating that no systematic biases seem to have occurred. By the end of the survey, 520 respondents provided valid answers, translating to an attrition rate of 13.3%.

Due to the anonymous and non-invasive nature of the research (being a voluntary, one-time participation of adults in survey-research on professional practice), no official approval by the ethics committee was sought. Following the European Code of Conduct for Research Integrity, all respondents agreed to a written informed consent stating that participation in the research was voluntary, that they could stop their participation at any moment, and that their answers would be processed anonymously.

### Instrument

**Item creation and selection.**   First, a pool of items was created by three researchers with expert knowledge on diversity in education. In order to maintain the goals of the DISCO-instrument and reflect the central tenets of EfA, the following criteria were observed during this creation phase: (1) to mitigate the fragmentation of the EfA field and employ a broad definition of diversity, items had to either be applicable to all children or refer to specific diversity groups in a balanced way; (2) items needed to explicitly measure a single practice or perception; and (3) items needed to be relevant across multiple teaching contexts; this implied that items had to be recognisable for teachers across different teaching levels, including primary and secondary education; that items had to refer to actions that could be performed by all

teachers, and not just those teaching specific courses or students; and had to refer to actions pertaining to day-to-day teaching activities in order to reflect the continuous nature of working on inclusion. For the wording of the items, the brainstorm was based on an existing reflection-instrument for how teachers address diversity in the classroom, and validated scales on ethnocentrism, homonegativity, differentiation, collaboration, and teaching efficacy. In order to ensure content validity and readability, these items were then adapted based on feedback by two academics in the field of diversity and (teacher) education, two in-service teacher educators specialized in teacher professional development for diversity, and two teachers. Through this process, the questionnaire ended up consisting of 128 items, which were then presented in the first step of the research to a large-scale group of pre-service teachers. The questionnaire was administered in Dutch, the official language of Flanders.

**The questionnaire.** The questionnaire consists of two parts. Part one considers teachers' beliefs by answering on a 7-point Likert scale (0 = *completely disagree*, 6 = *completely agree*) to allow for nuanced answers, in line with psychometric recommendations for Likert-scales [49]. Furthermore, given that inclusive teaching has become an obligatory professional competence, and hence a norm in education, this increased the chance of social desirability bias and ceiling effects in respondents' answers [50]. To counteract this, we provided respondents with a large range of items that were critical of inclusion or student diversity, and hence, were reverse-coded. The first component taps teachers' professional beliefs on diversity, specifically with regards to students with ethnic minority background (e.g., "The more immigrant students in a school, the more discipline problems the school will face"), with a disability (e.g., "Students with a disability reduce the learning opportunities for the other students in the class"), from lower socio-economic backgrounds (e.g., "underprivileged parents are often less interested in their children's progress in school than other parents"), with regards to essentialist notions on the gender binary (e.g., "I think it's good that girls are not asked to help out with jobs in school that involve heavy lifting"), and LGB-students (e.g., "many students experiment with their sexuality just to get noticed").

The second component consists of responsibility-beliefs, and is divided in two sections. The first one taps the extent to which teachers believe it is the responsibility of the educational field to create maximal developmental opportunities for diverse students. These 33 items consider general items (e.g., "Teachers have a didactical task in the classroom, not one of raising students or caring for them.") as well as with regards to specific groups of students (e.g., "The school must allow students with a migration background to speak a language other than Dutch at school", "it is time to pay attention to LGB-sexuality in several parts of the curriculum"). The second section taps teachers' perceptions of to what extent they should collaborate with both diverse parents and colleagues. These 16 items are preceded by the stem "It is my job to..." followed by, for instance, "Establish a good relationship with all parents, even if I do not like them" or "learn from colleagues with a completely different teaching approach".

The second part of the questionnaire contains 42 items considering self-efficacy, consisting of three dimensions. In line with other validated efficacy scales [51,52], these items are preceded by the stem "To what extent are you able to..." and is answered on a 9-point Likert scale (0 = not at all, 8 = a lot). The first dimension contains 9 items tapping teachers' perceptions of being able to notice student diversity (f.i., students' background, interests, well-being, ...) with, for instance, "make time for in-depth individual conversations with all of your students in order to provide more effective support". The second dimension taps the extent to which teachers feel capable to design a classroom environment that takes into account the needs of their students and contains 21 items (e.g., "Give students opportunities beyond traditional tests and exams to demonstrate their knowledge and skills"). The third dimension measures the extent to which teachers feel capable to create positive interactions among students through 12 items (e.g., "Coach students to treat other opinions with respect.").

**Plan of analysis.** To validate the factor structure underlying the DISCO-instrument, we combined exploratory (EFA) and confirmatory factor analyses (CFA) in a stepwise approach. In order to maximize power for the analyses and following the full information maximum likelihood (FIML) on missing data theory [53], all available data was used for the factor analyses. Hence, respondents were not a priori list-wise deleted from the dataset for having missings on some survey-items. In the first step, using the data from pre-service teachers, exploratory factor analysis using SPSS 25 was chosen to identify a (set of) latent constructs underlying the measured items without strong theoretical assumptions on how many factors existed [54]. Because teachers' competences in handling diversity were expected to correlate [54,55], Principal Component Analysis (PCA) with oblique rotation technique (direct oblimin) was used. The assumptions to perform EFA were checked: data distributions were checked to ensure that variables were roughly normally distributed [54] and the correlation-matrix was inspected to identify scale items that correlated below |.30| with all other variables or above |.80|. The suitability of the data for the exploratory analysis were checked by considering the Kaiser–Meyer–Olkin test (KMO) which should exceed the recommended value of .60, and the Bartlett's test of sphericity which should be significant. Furthermore, the anti-image correlation matrix was checked for negative correlations, low communalities or high partial correlations [54]. To determine the number of factors, both the scree plots' point of inflexion and Kaiser's criterion (eigenvalue >1) were checked [54]. Items were eliminated if they had inadequate (partial) correlations or high cross-loadings (>|.40|), if their variance was not well explained as suggested by low communalities, or if the loading on their designated factor was weak (<|.35|). When a choice for being dropped or retained had to be made between items, the theoretical interpretability and alignment with EfA tenets spurred the decision besides psychometric considerations. Factors with a minimum of three items and Cronbach's alpha >.70 were retained for testing in the second step [54].

For the second step, the survey was adapted according to the insights from the first step and then presented to a new sample consisting of in-service teachers. The initial factor solution from step 1 was checked with data from these in-service teachers using confirmatory factor analyses in Mplus. This technique allows us to check to what extent a hypothesized model fits the data. An MLR estimation was used, and goodness of fit indicators were checked: CFI values larger than 0.95, $X^2$/DF smaller than two, RMSEA values smaller than 0.05 and SRMR values smaller than 0.05 are considered good [56]. Where necessary, the model was optimized based on suggestions of the modification indices, standardized factor loadings, R-square estimates of the items, and theoretical interpretability of the factor. To compare models, we checked which model had a lower AIC value. Note that $X^2$ statistics are reported to increase transparency for the $X^2$/DF calculation, but that the $X^2$ statistics themselves are not discussed because the test is sensitive to large sample sizes [57].

Finally, based on these results, the scale scores for in-service teachers were obtained by computing a mean for each respondent with a maximum of 25% missing values on the scale-items. The final set of items for each scale can be seen in Tables 2 and 3. Repeated measures analysis of variance with a bonferroni post-hoc test was conducted to examine possible significant differences on scale scores for efficacy and beliefs.

## Results

### Step 1: EFA with pre-service teachers

**Professional beliefs about diverse students.** Based on the data distributions, correlation matrix and anti-image correlation matrix, ten items were deleted (see Table 1). PCA on the remaining 27 items revealed satisfactory KMO-value, anti-image correlations and Bartletts'

**Table 1. Overview of decisions on all scale-items and factors throughout the analyses (Note that items relating to the factor "broad role of teachers" are indicated in grey for recognazibility).**

| original scale-name | | Item Nr | To what extent are you able to. . .? / Do you agree with . . .? | Decision after step 1 | Reason for decision | Decision after step 2 | Reason for decision | Final scale-name | |
|---|---|---|---|---|---|---|---|---|---|
| Self-efficacy in | Noticing diversity | C1.01 | Talk to colleagues to evaluate or adjust your image of a student | delete | low communality score | | | Noticing diversity | Self-efficacy in |
| | | C1.02 | Make time for in-depth individual conversations with all of your students in order to provide more effective support. | | | | | | |
| | | C1.03 | Gain insight into the learning needs of an individual student by consciously looking at how he/she responds to different tasks and works in different groupings. | | | | | | |
| | | C1.04 | Gain insight into a student's skills and learning tempo by checking their independent work while they are working on it. | delete | high partial correlations with C1.03 | | | | |
| | | C1.05 | Gain insight into a student's plans and dreams for the future. | | | | | | |
| | | C1.06 | Start a conversation with a student who is not attentive or is having a difficult time outside of the classroom. | | | | | | |
| | | C1.07 | Gain insight into social relationships among students. | | | | | | |
| | | C1.08 | Get a complete picture of a student during the first lessons (to a class group). | delete | negative correlations with all other items | | | | |
| | | C1.09 | Gain insight into the students' feelings about their family situation. | | | | | | |
| | High-quality student-interactions | C3.01 | Have students work together in heterogeneous groups. | delete | low communality score | | | High-quality student-interactions | |
| | | C3.02 | Rrespond to racist statements by students. | delete | skewed distribution / ceiling effect | | | | |
| | | C3.03 | Allow students with learning difficulties to be helped by other pupils. | | | | | | |
| | | C3.04 | Coach students to treat other opinions with respect. | delete | skewed distribution / ceiling effect | | | | |
| | | C3.05 | Give tasks where the students have to work together to complete the task successfully (for example, each having a different role). | | | | | | |
| | | C3.06 | Provide feedback on how students work together. | | | | | | |
| | | C3.07 | Create space in or outside the classroom to support students in the resolution of conflicts. | | | | | | |
| | | C3.08 | Respond to homophobic statements made by students. | delete | skewed distribution / ceiling effect | | | | |
| | | C3.09 | Coach students to give each other thorough and respectful feedback. | | | delete | suggested by modification indices | | |
| | | C3.10 | In group work, ensure that all pupils get to work with others from all across the class? | | | | | | |
| | | C3.11 | draw up clear rules for the classroom together with the pupils. | | | delete | suggested by modification indices | | |
| | | C3.12 | In the case of bullying, discuss a plan of action with the students involved. | | | | | | |
| | Creating stimulating learning environment | C4.01 | Ensure that teaching materials reflect diversity in society. | | | | | Creating stimulating learning environment | |
| | | C4.02 | Break stereotypes. | delete | high partial correlations with C4.01 | | | | |
| | | C4.03 | Address sensitive themes, such as discrimination, radicalization, gaybashing, cat-calling,. . . | delete | inadequate score on anti-image correlation | | | | |
| | | C4.04 | Use a wide range of evaluation methods in order to evaluate students in a broad a way as possible. | | | | | | |
| | | C4.05 | Take into account how much progress students have made when I evaluate them. | delete | high partial correlations with C4.04 | | | | |
| | | C4.06 | Specifically check that students with a language delay have understood a question. | | | | | | |
| | | C4.07 | Enter into dialogue with students about their results. | | | | | | |
| | | C4.08 | Occasionally use a short moment of evaluation (e.g., quiz) to tailor your lesson in a more nuanced way to the students' needs. | | | | | | |
| | | C4.09 | take the linguistic background of students into account during evaluation moments (e.g. by not counting language errors). | delete | low communality score | | | | |
| | | C4.10 | Give students opportunities beyond traditional tests and exams to demonstrate their knowledge and skills (e.g. through portfolios, writing tasks, oral presentations). | | | | | | |
| | | C4.11 | Allow students to give input about content that they would like to cover in lessons. | | | | | | |
| | | C4.12 | Have students evaluate your lessons. | | | | | | |
| | | C4.13 | When managing the pace of your lessons, take account of students who work faster/slower than the others. | | | | | | |
| | | C4.14 | Offer a variety of activities so that students can choose what to do themselves. | | | | | | |
| | | C4.15 | Provide a wide variety of assignments, according to the needs and skill level of students. | delete | high partial correlations with C4.14 | | | | |
| | | C4.16 | Allow students to choose whether or not to use certain tools/support. | | | | | | |
| | | C4.17 | Present content in different ways (eg text, audio material, visual material). | delete | high partial correlations with C4.18 | | | | |
| | | C4.18 | Use a variety of teaching materials and media which invite pupils to draw on different senses. | | | | | | |
| | | C4.19 | When necessary, adapt your learning objectives (knowledge and skills) to take account of differences between students (e.g., by creating main and advanced learning goals). | | | | | | |
| | | C4.20 | Allow all students to 'show off' sufficiently with what they are good at, including their extracurricular skills. | delete | inadequate score on anti-image correlation | | | | |
| | | C4.21 | Let all students 'show off' sufficiently with what they have done during an exercise, homework, class assignment, role play. | delete | high partial correlations with C4.20 | | | | |

*(Continued)*

**Table 1.** (Continued)

| original scale-name | | Item Nr | To what extent are you able to. . .? / Do you agree with . . .? | Decision after step 1 | Reason for decision | Decision after step 2 | Reason for decision | Final scale-name |
|---|---|---|---|---|---|---|---|---|
| Beliefs on the responsability of the educational field to create EfA, focused on | collaborating with parents & colleagues | C5.01 | Make parents from different backgrounds feel comfortable about coming into school. | stem changed from "it is my job to. . ." to "To what extent are you able to. . .?" | skewed distribution / ceiling effect | | | collaborating with parents |
| | | C5.02 | Help parents to support their children to do well in school. | | | | | |
| | | C5.03 | Involve the parents of students with special needs in their children's educational career. | | | | | |
| | | C5.04 | Adapt the way you communicate with different kinds of families (e.g. single parents, LGBT+ parents, non-native speakers. . .). | | | | | |
| | | C5.05 | Empathize with the situation and motivations of different parents. | | | delete | suggested by modification indices | |
| | | C5.06 | Use the insights that parents give you about their child. | | | | | |
| | | C5.07 | Establish a good relationship with all parents, even if you do not like them. | | | | | |
| | | C5.08 | Actively make time for parents who want to have a conversation with me. | | | delete | suggested by modification indices | |
| | | C5.14 | Always be available for questions from parents during school hours. | | | | | |
| | | C5.15 | Allow parents to give some input about my lessons. | | | | | |
| | | C5.09 | Consult other professionals (e.g.: SEN teachers, speech therapists, . . .) when one of your students has extra needs. | | | delete | suggested by theoretical interpretability, r-square values and factor loadings | collaborating with colleagues |
| | | C5.10 | Involve the school team (e.g.: teachers, student counselor, . . .) when one of your students has extra needs. | | | delete | suggested by modification indices, r-square values and factor loading | |
| | | C5.11 | Integrate lesson ideas from colleagues into your own teaching practice. | | | | | |
| | | C5.12 | Ask colleagues for feedback to improve your teaching. | | | | | |
| | | C5.13 | Create new teaching methods or materials together with colleagues in order to support students with learning difficulties. | | | | | |
| | | C5.16 | Share learning materials with colleagues that meet students' different learning profiles. | | | delete | suggested by r-square values and factor loadings | |
| | Ethnic minorities | C6.01 | The school should provide specific services for students who want to follow religious practices (e.g., prayer room, halal/kosher food). | | | | | Ethnic minorities | Beliefs on the responsability of the educational field to create EfA, focused on |
| | | C6.02 | The school must allow students with a migration background to speak a language other than Dutch at school. | | | | | |
| | | C6.03 | As a school, you should not take into account the fatigue of Muslim students participating in Ramadan. | delete | low correlations with other items | | | |
| | | C6.04 | The school library should have books in the different home languages of the students. | | | | | |
| | | C6.05 | It is important to encourage students of immigrant origin to be proud of their ethnic and cultural background. | | | | | |
| | | C6.06 | As a teacher, it is your job to react to racist statements by students. | delete | low correlations with other items | | | |
| | Disability | C6.07 | It is time that we give children with a disability a full place in mainstream education. It is their right. | | | | | Disability |
| | | C6.08 | It is sensible to not send students to special educational support too quickly. | | | | | |
| | | C6.09 | The school's infrastructure must be made accessible at all levels to people with physical and sensory disabilities (vision, hearing, motor). | | | | | |
| | | C6.10 | You cannot expect the teacher to meet the needs of a child with a disability in mainstream education. | | | | | |
| | | C6.11 | It is normal to prepare extra tasks for students who work at a higher or lower level. | delete | low correlations with other items | | | |
| | SES | C6.12 | The influence of a teacher is limited compared to the influence of the pupil's home environment. | delete | part of scale with low cronbach's alpha | | | General policy |
| | | C6.13 | Teachers must be capable of working with students from different home situations. | | | | | |
| | | C6.14 | Students from families with financial problems should be provided with appropriate support (e.g. payment plan, special support fund for students who cannot pay the school fees). | | | | | |
| | | C6.15 | A school cannot be held responsible if underprivileged pupils benefit little from the education offered. | delete | low correlations with other items | | | |
| | | C6.16 | A teacher can make a difference for students from all kinds of home environment. | | | | | |
| | Gender binary | C6.17 | You cannot expect a school to change the toilet infrastructure for a single transgender student. | delete | low correlations with other items | | | |
| | | C6.18 | Students sending sexually suggestive photos to each other is beyond the responsibility of the school. | delete | low correlations with other items | | | |
| | | C6.19 | Making students aware of inequality between men and women is outdated. | delete | low correlations with other items | | | |
| | | C6.20 | It is important that we as a school respond quickly and consistently to incidents of sexual harassment among students. | delete | skewed distribution / ceiling effect | | | |
| | | C6.21 | As a school, it is important to break the traditional divide between subjects that are stereotypically for boys or for girls. | | | | | |
| | LGB | C6.22 | As a teacher, it is your job to respond to homophobic statements by students. | | | delete | suggested by r-square values and factor loading | |
| | | C6.23 | It is too cumbersome to adapt all school communication (f.i. the salutation in a letter) to LGB parents. | delete | low correlations with other items | | | |
| | | C6.24 | It is time that we have attention for LGB sexuality in various parts of the curriculum. | delete | high partial correlations with C6.25 | | | |
| | | C6.25 | It is our job as a school to support students coming out of the closet during this process. | delete | low communality score | | | |
| | | C6.26 | As a school, It is normal to support LGB colleagues through thick and thin, even in the event of negative reactions from the school community. | delete | skewed distribution / ceiling effect | | | |
| | A general approach | C6.27 | It is the responsibility of every teacher to support the language development of pupils | delete | skewed distribution / ceiling effect | | | |
| | | C6.28 | Teachers have a didactical task in the classroom, not one of raising students or caring for them. | delete | part of scale with low cronbach's alpha | | | |
| | | C6.29 | It is important that your work contributes to the personal growth of students. | | | delete | suggested by modification indices and low factor loading | |
| | | C6.30 | For a teacher, professional knowledge is more important than knowledge of didactics and psychology. | delete | part of scale with low cronbach's alpha | | | |
| | | C6.31 | The future of the current generation of students often concerns me | delete | low communality score | | | |
| | | C6.32 | I think it is important to mean something personally to students. | delete | high partial correlations with C6.29 | | | |
| | | C6.33 | A teacher is not a youth or social worker. | delete | part of scale with low cronbach's alpha | | | |
| | | C6.43 | A school has the task of combating inequality in society | new item added | | delete | suggested by modification indices | |
| | | C6.44 | It is every teacher's job to contribute to a school policy that is mindful of diversity. | new item added | | | | |

(Continued)

**Table 1.** (Continued)

| original scale-name | | Item Nr | To what extent are you able to. . .? / Do you agree with . . .? | Decision after step 1 | Reason for decision | Decision after step 2 | Reason for decision | Final scale-name | |
|---|---|---|---|---|---|---|---|---|---|
| professional beliefs on student diversity | Ethnic minorities | C2.01 | Ethnic and cultural diversity is a valuable resource in education. | | | | | Ethnic minorities | professional beliefs on student diversity |
| | | C2.02 | Immigrant parents are less involved in their child's school career. | delete | factor-loading <0.3 | | | | |
| | | C2.03 | If there are more "native-born" pupils in a school, it means that academic achievement will be higher across the board. | | | | | | |
| | | C2.04 | If an additional language is spoken at home, the student will fall behind in their learning. | | | | | | |
| | | C2.05 | Schools that educate many migrant students have lower standards. | delete | high partial correlations with C2.35 | | | | |
| | | C2.06 | Immigrant students are not as good at mathematics as native students. | | | | | | |
| | | C2.07 | Students who speak a different language at home will never speak Dutch properly. | | | | | | |
| | | C2.08 | The more immigrant students in a school, the more discipline problems the school will face. | | | | | | |
| | LGB | C2.09 | It doesn't matter to me if my principal has a same-sex partner. | delete | skewed distribution / ceiling effect | | | essentialist notions on the gender binary & sexual orientation | |
| | | C2.10 | It is better for partners of LGBs not to come to school parties and other school activities. | delete | skewed distribution / ceiling effect | | | | |
| | | C2.11 | Many students experiment with their sexuality just to get noticed. | | | | | | |
| | | C2.12 | LGB students who behave in an exaggerated manner are asking to be bullied. | | | delete | suggested by modification indices | | |
| | | C2.13 | LGB teachers have the right to be open with students about their sexual orientation. | delete | multicollinearity with C2.14 | | | | |
| | | C2.14 | LGB teachers (lesbian, gay, bisexual) have the right to be open with parents about their sexual orientation. | | | | | | |
| | | C2.15 | In order for their time in school to go smoothly, it is better for LGBT students not to openly display their sexual orientation. | | | | | | |
| | | C2.16 | I feel sorry for students who grow up in an LGBT family because it is not easy for them. | | | | | | |
| | Gender binary | C2.25 | Boys shouldn't wear earrings or nail varnish when in school. | | | | | | |
| | | C2.26 | It is no problem at all that girls opt for 'soft' study options/professions (e.g. languages, care) and boys for 'hard' study options/professions (e.g. sciences, technology). | delete | low correlations with other items | | | | |
| | | C2.27 | Two boys should be able to dance together for the entire evening at the school dance without attracting undue attention. | | | | | | |
| | | C2.28 | I find it difficult to understand transgender students. | | | | | | |
| | | C2.29 | I think it's good that girls are not asked to help out with jobs in school that involve heavy lifting. | | | | | | |
| | | C2.30 | Girls will generally work more neatly and punctually than boys. | delete | low correlations with other items | | | | |
| | | C2.31 | In a class with many boys, classroom management is often more difficult. | delete | low correlations with other items | | | | |
| | Disability | C2.17 | The care for pupils with a disability is worthwhile, but is often at the expense of attention for other pupils. | | | delete | suggested by modification indices, r-square values and factor loading | Disability | |
| | | C2.18 | I find it difficult to like students with behavioral disorders as much as other students. | | | delete | suggested by r-square values and factor loading | | |
| | | C2.19 | I find it difficult to teach students with a learning disability. | delete | high partial correlations with C2.18 | | | | |
| | | C2.20 | Students with a disability always score worse on tests than other students. | | | | | | |
| | | C2.21 | Students with a disability reduce the learning opportunities for the other students in the class. | | | | | | |
| | | C2.22 | Students with a disability often abuse the support that they receive. | | | | | | |
| | | C2.23 | It is fair that a stronger student is given a more difficult test. | delete | low correlations with other items | | | | |
| | | C2.24 | I think that many teachers show too much understanding for the behaviour of students with a disability. | | | | | | |
| | SES | C2.32 | How many abilities a pupil has is independent of the home environment. | delete | low correlations with other items | | | SES | |
| | | C2.33 | Underprivileged parents are often less interested in their children's progress in school than other parents. | | | | | | |
| | | C2.34 | It is difficult to be sympathetic towards poorer students if they or their parents always have the latest mobile phones or gadgets. | | | | | | |
| | | C2.35 | Schools that educate many underprivileged students have lower standards. | | | | | | |
| | | C2.36 | Underprivileged students rarely succeed in difficult subjects because they do not have supportive parents. | | | | | | |
| | | C2.37 | It is acceptable for teachers to have higher expectations of students from well-off backgrounds. | | | | | | |

test. The scree plot and Kaiser criterion indicated that it was reasonable to retain four factors. Inspection of the rotated solution revealed an item loading below .3 unto its designated factor, which was deleted. The PCA on the remaining 26 items similarly revealed that it was reasonable to retain four factors. Communalities ranged from .32 to .60. These four factors explained 46.48% of the total variance and had eigenvalues of 25.81, 9.83, 6.13, and 4.71 respectively. To interpret these factors, we made use of an oblimin rotation. The factor correlation matrix indeed shows that there was a moderate correlation between the various factors (r-values between |.152| and |.363|). The first factor consisted of 6 items tapping beliefs towards ethnic minority students, with factor loadings ranging from .38 to .77, and a Cronbach's alpha of .77. The second factor consisted of 9 items tapping beliefs on sexual orientation and essentialist notions on the gender binary, with factor loadings ranging from .51 to .69, and a Cronbach's alpha of .82. The third factor consisted of 6 items tapping beliefs towards students with a

disability, with factor loadings ranging from .57 to .72, and a Cronbach's alpha of .77. The final factor consisted of 5 items tapping beliefs towards low SES-students, with factor loadings ranging from .42 to .72, and a Cronbach's alpha of .72.

**Responsibility of the educational field to create Education for All.** Based on the data distributions, correlation matrix, and anti-image correlation matrix, fifteen items were deleted (see Table 1). PCA on the remaining 18 items revealed satisfactory KMO-value, anti-image correlations and Bartletts' test, and that it was reasonable to retain four factors. Communalities ranged from .41 to .72. These four factors explained 53.98% of the total variance and had eigenvalues of 25.42, 11.55, 9.04, and 7.97 respectively. The factor correlation matrix shows that there was a correlation between the various factors (r-values between |.134| and |.281|). Interestingly, the revealed factors did not line up directly with the diversity groups included in the items. Rather, the first factor, consisting of 6 items, seemed to tap a broad sense of school responsibility, including general items and items with regards to tackling discrimination based on SES, gender and sexuality. This factor had loadings ranging from .62 to .75, and a Cronbach's alpha of .71. The second factor tapped beliefs on school approaches specifically directed at ethnic minority students, consisting of 4 items with loadings ranging from .45 to .86, and a Cronbach's alpha of .70. The third factor consisted of 4 items representing beliefs on the broad role teachers should take on, with factor loadings ranging from .59 to .71, and had a Cronbach's alpha of .61. The final factor consisted of 4 items tapping beliefs on schools' responsibilities towards students with a disability, with factor loadings ranging from .64 to .83, and had a Cronbach's alpha of .75.

In acknowledgement of this rather surprising factorial structure, we decided for the second step to add a few more items representing a general sense of schools' responsibilities towards all diverse students, rather than geared towards specific groups. However, the items pertaining to the factor representing teachers' broad role were dropped from the questionnaire for the second step, due to its low Cronbach's alpha.

**Responsibility beliefs towards diverse parents and colleagues.** Inspection of the data distributions revealed that respondents had answered the items in a rather extreme manner. That is, the median-likert category was at least 5 = *agree* for all items, indicating that 50% of respondents scored 5 or higher on scales going from 0 to 6. As this indicated severe ceiling-effects, it was decided to rework the dimension as a whole before submitting the questionnaire to in-service teachers in step 2. It was hypothesized that the stem "It is my job to. . ." was leading to a certain social desirability bias rather than providing a reflection of actual intentions. Consequently, it was deemed advisable to rework the scale to tap respondents' self-efficacy in dealing with diverse parents and colleagues rather than their responsibility beliefs. Hence, in-service teachers were presented with 16 items preceded by the stem "To what extent are you able to. . ." and a 9-point Likert scale (0 = *not at all*, 8 = *a lot*).

**Self-efficacy in noticing student diversity.** The 9 items had a KMO value of .902 and Bartletts' test was significant ($X^2$ = 4584, df = 36, $p < .001$). Inspecting the anti-image correlation matrix led to the deletion of three items (see Table 1). PCA on the remaining 6 items revealed satisfactory KMO-value, anti-image correlations and Bartletts' test. The scree plot and Kaiser criterion indicated one factor to extract with an eigenvalue 4.00, which explained 67% of the total variance. Factor loadings ranged from .76 to .84, and communalities ranged from .58 to .72. The scale had high internal reliability ($\alpha$ = .90).

**Self-efficacy in creating stimulating learning environments for diverse students.** Inspection of the correlation matrix and anti-image correlation matrix revealed several problematic item-sets, so eight items were deleted (see Table 1). PCA on the remaining 13 items revealed satisfactory KMO-value, anti-image correlations and Bartletts' test. The scree plot and Kaiser criterion indicated one factor to extract with an eigenvalue 7.43, which explained 57.2%

of the total variance. Factor loadings ranged from .70 to .80, and communalities ranged from .48 to .63. The scale had high internal reliability ($\alpha$ = .94).

**Self-efficacy in creating high-quality student-interactions.**    Based on the data distributions and anti-image correlation matrix, four items were deleted (see Table 1). PCA on the remaining 8 items revealed satisfactory KMO-value, anti-image correlations and Bartletts' test. The scree plot and Kaiser criterion indicated one factor to extract with an eigenvalue 5.18, which explained 64.7% of the total variance. Factor loadings ranged from .70 to .85, and communalities ranged from .49 to .72. The scale had high internal reliability ($\alpha$ = .92).

**Step 2: CFA with in-service teachers.**    In the second step, the solution that resulted from the first step was checked for in-service teachers using CFA. With regards to self-efficacy, we included four different factors: noticing diversity with six items, enabling high-quality student-interactions with eight items, creating stimulating learning environments with 13 items, and collaborating with colleagues and parents with 16 items. With regard to beliefs about diverse students, we included four factors geared toward specific groups, being ethnic minority students, disabled students, students from low SES, and essentialist notions regarding the gender binary and sexuality with respectively six, six, five, and nine items. For beliefs on the responsibility of the educational field to create education for all, we included three factors focusing on general school policy, and initiatives geared specifically towards either ethnic minority students or students with a disability, with respectively eight, four and four items. The result of this points to acceptable but not good fit indices (AIC = 155650, $X^2$ = 8030, DF = 3430, p < 0.001, $X^2$/DF = 2.34, RMSEA = 0.048, CFI = 0.760, SRMR = 0.066). Changes were made in order to achieve a better fit. As factor loadings for the items related to collaborating with colleagues were consistently lower (i.e., between .424 and .511), we tested a model with a separate factor pertaining to collaborating with diverse parents (consisting of 10 items) and one pertaining to collaborating with colleagues (consisting of six items). These adaptations led to a better fit (AIC = 154671, $X^2$ = 7125, DF = 3419, p < 0.001, $X^2$/DF = 2.08, RMSEA = 0.043, CFI = 0.807, SRMR = 0.059). In a next step, 13 items were deleted as suggested by the modification indices, R-square values or factor loadings (see Table 1). This led to a further improved model (AIC = 133214, $X^2$ = 4765, DF = 2418, p < 0.001, $X^2$/DF = 1.97, RMSEA = 0.041, CFI = 0.846, SRMR = 0.054). In accordance with suggestions by Hu and Bentler (1999) and as shown by Coubergs and colleagues (2017), we accept a combination of CFI-values approaching .90 with RMSEA values smaller than .06 and SRMR-values under .08 as good model fit. Factor loadings can be seen in Tables 2 and 3 (note that the loadings for self-efficacy and beliefs are shown in separate tables for lay-out reasons only, as these are tested as one model within CFA). This table shows loadings above |.50| for all items, with the exception of C2_14, C2_27, C2_29, C5_14, and C5_15. As these items still loaded above |.40|, while deleting them was not conducive to the Cronbach's alpha of the respective factor, this was deemed acceptable. All in all, this results in a model with five factors related to self-efficacy and seven factors related to beliefs, with alpha scores being good for each factor and showing that no items could be deleted in order to get a higher alpha. More specifically, the factors related to self-efficacy comprises "noticing student diversity" consisting of 6 items with $\alpha$ = .87, "enabling high-quality student-interactions" consisting of 6 items with $\alpha$ = .85, "creating stimulating learning environments" consisting of 13 items with $\alpha$ = .90, "collaborating with colleagues" consisting of 3 items with $\alpha$ = .83, and "collaborating with parents from diverse backgrounds" consisting of 8 items with $\alpha$ = .88. The factors relating to beliefs comprises professional beliefs on different types of diversity, specifically on ethnic minorities (# of items = 6, $\alpha$ = .79), students with a disability (# = 4, $\alpha$ = .78), students from low-SES families (# = 5, $\alpha$ = .76), and essentialist notions on sexuality and the gender binary (# = 8, $\alpha$ = .79), as well as beliefs on the responsibility of the educational field to create education for all, including a

**Table 2. Standardised CFA matrix for self-efficacy on handling diversity.**

| | To what extent are you able to. . .? | Notice diversity | High-quality student-interactions | Stimulating learning environment | collaborate with parents | collaborate with colleagues |
|---|---|---|---|---|---|---|
| C1.02 | Make time for in-depth individual conversations with all of your students in order to provide more effective support. | 0.763 | | | | |
| C1.03 | Gain insight into the learning needs of an individual student by consciously looking at how he/she responds to different tasks and works in different groupings. | 0.686 | | | | |
| C1.05 | Gain insight into a student's plans and dreams for the future. | 0.674 | | | | |
| C1.06 | Start a conversation with a student who is not attentive or is having a difficult time outside of the classroom. | 0.688 | | | | |
| C1.07 | Gain insight into social relationships among students. | 0.775 | | | | |
| C1.09 | Gain insight into the students' feelings about their family situation. | 0.817 | | | | |
| C3.03 | Allow students with learning difficulties to be helped by other pupils. | | 0.657 | | | |
| C3.05 | Give tasks where the students have to work together to complete the task successfully (for example, each having a different role). | | 0.619 | | | |
| C3.06 | Provide feedback on how students work together. | | 0.765 | | | |
| C3.07 | Create space in or outside the classroom to support students in the resolution of conflicts. | | 0.761 | | | |
| C3.10 | In group work, ensure that all pupils get to work with others from all across the class? | | 0.678 | | | |
| C3.12 | In the case of bullying, discuss a plan of action with the students involved. | | 0.710 | | | |
| C4.01 | Ensure that teaching materials reflect diversity in society. | | | 0.580 | | |
| C4.04 | Use a wide range of evaluation methods in order to evaluate students in a broad a way as possible. | | | 0.697 | | |
| C4.06 | Specifically check that students with a language delay have understood a question. | | | 0.693 | | |
| C4.07 | Enter into dialogue with students about their results. | | | 0.547 | | |
| C4.08 | Occasionally use a short moment of evaluation (e.g., quiz) to tailor your lesson in a more nuanced way to the students' needs. | | | 0.693 | | |
| C4.10 | Give students opportunities beyond traditional tests and exams to demonstrate their knowledge and skills (e.g. through portfolios, writing tasks, oral presentations). | | | 0.613 | | |
| C4.11 | Allow students to give input about content that they would like to cover in lessons. | | | 0.685 | | |
| C4.12 | Have students evaluate your lessons. | | | 0.537 | | |
| C4.13 | When managing the pace of your lessons, take account of students who work faster/slower than the others. | | | 0.731 | | |
| C4.14 | Offer a variety of activities so that students can choose what to do themselves. | | | 0.644 | | |
| C4.16 | Allow students to choose whether or not to use certain tools/support. | | | 0.719 | | |
| C4.18 | Use a variety of teaching materials and media which invite pupils to draw on different senses. | | | 0.686 | | |
| C4.19 | When necessary, adapt your learning objectives (knowledge and skills) to take account of differences between students (e.g., by creating main and advanced learning goals). | | | 0.678 | | |
| C5.01 | Make parents from different backgrounds feel comfortable about coming into school. | | | | 0.819 | |
| C5.02 | Help parents to support their children to do well in school. | | | | 0.870 | |

(*Continued*)

**Table 2.** (Continued)

| To what extent are you able to. . .? | | Notice diversity | High-quality student-interactions | Stimulating learning environment | collaborate with parents | collaborate with colleagues |
|---|---|---|---|---|---|---|
| C5.03 | Involve the parents of students with special needs in their children's educational career. | | | | 0.885 | |
| C5.04 | Adapt the way you communicate with different kinds of families (e.g. single parents, LGBT+ parents, non-native speakers. . .). | | | | 0.760 | |
| C5.06 | Use the insights that parents give you about their child. | | | | 0.706 | |
| C5.07 | Establish a good relationship with all parents, even if you do not like them. | | | | 0.720 | |
| C5.14 | Always be available for questions from parents during school hours. | | | | 0.480 | |
| C5.15 | Allow parents to give some input about my lessons. | | | | 0.451 | |
| C5.11 | Integrate lesson ideas from colleagues into your own teaching practice. | | | | | 0.769 |
| C5.12 | Ask colleagues for feedback to improve your teaching. | | | | | 0.801 |
| C5.13 | Create new teaching methods or materials together with colleagues in order to support students with learning difficulties. | | | | | 0.800 |

factor on general school policy (# = 5, $\alpha$ = .74), and factors with initiatives geared specifically towards ethnic minority students (# = 4, $\alpha$ = .70), and students with a disability (# = 4, $\alpha$ = .73).

**Scale-scores.** With regard to beliefs, analysis of variance showed that there were significant differences on in-service teachers' belief scores depending on the dimension ($V$ = .825, F(6, 514) = 101.26, p < .001). More specifically, teachers' scores varied on average between 3 and 5 on scales ranging from 0 to 6 (see Table 4), indicating that teachers' beliefs varied from a neutral position to "agree" depending on the specific scale. The post-hoc Bonferroni test revealed that, with the exception of teachers' beliefs on SES students and their beliefs on school policy specifically geared towards disability students (p < .887), all scale scores differed significantly from each other. Hence, teachers scored highest on their belief on schools' responsibility to have a general policy on diversity, followed by holding most favourable beliefs with regards to students' gender and sexuality, and students with a disability, followed by ethnic minority students. Teachers held the least positive beliefs on the responsibility of a school to have a policy specifically geared towards ethnic minority students, and scored somewhat higher on their beliefs on SES students and the responsibility of a school to have a policy specifically geared towards disability, which did not differ significantly (as reported above).

As shown in Table 4, in-service teachers tended to score on average between 5 and 6 on the efficacy-scales ranging from 0 to 8, indicating that teachers feel more or less "reasonably" efficacious. Analysis of variance showed that there were significant differences on efficacy scores depending on the dimension ($V$ = .438, F(4, 520) = 101.26, p < .001). The post-hoc Bonferroni test revealed that teachers did not differ significantly in their efficacy in enabling high-quality interactions between students and collaborating with colleagues. These were the dimensions on which teachers tended to feel most efficacious when compared to the other dimensions (p < .001), with creating stimulating learning environments taking a middle position (p < .01), while noticing student diversity and collaborating with parents did not differ significantly from each other and were the dimensions that teachers rated themselves least efficacious on (p < .01).

**Table 3. Standardised CFA matrix for beliefs in handling diversity.**

| | | Beliefs on different types of diversity | | | | Schools' responsibility for EfA | | |
|---|---|---|---|---|---|---|---|---|
| **To what extent do you agree with . . .?** | | **Ethnic minorities** | **Gender** | **Disability** | **SES** | **Ethnic minorities** | **Disability** | **General policy** |
| C2.01 | Ethnic and cultural diversity is a valuable resource in education. | 0.622 | | | | | | |
| C2.03 | If there are more "native-born" pupils in a school, it means that academic achievement will be higher across the board. | 0.451 | | | | | | |
| C2.04 | If an additional language is spoken at home, the student will fall behind in their learning. | 0.608 | | | | | | |
| C2.06 | Immigrant students are not as good at mathematics as native students. | 0.635 | | | | | | |
| C2.07 | Students who speak a different language at home will never speak Dutch properly. | 0.647 | | | | | | |
| C2.08 | The more immigrant students in a school, the more discipline problems the school will face. | 0.725 | | | | | | |
| C2.11 | Many students experiment with their sexuality just to get noticed. | | 0.557 | | | | | |
| C2.14 | LGB teachers (lesbian, gay, bisexual, transgender), have the right to be open with parents about their sexual orientation. | | 0.421 | | | | | |
| C2.15 | In order for their time in school to go smoothly, it is better for LGB students not to openly display their sexual orientation. | | 0.709 | | | | | |
| C2.16 | I feel sorry for students who grow up in an LGB family because it is not easy for them. | | 0.657 | | | | | |
| C2.25 | Boys shouldn't wear earrings or nail varnish when in school. | | 0.573 | | | | | |
| C2.27 | Two boys should be able to dance together for the entire evening at the school dance without attracting undue attention. | | 0.477 | | | | | |
| C2.28 | I find it difficult to understand transgender students. | | 0.675 | | | | | |
| C2.29 | I think it's good that girls are not asked to help out with jobs in school that involve heavy lifting. | | 0.479 | | | | | |
| C2.20 | Students with a disability always score worse on tests than other students. | | | 0.618 | | | | |
| C2.21 | Students with a disability reduce the learning opportunities for the other students in the class. | | | 0.681 | | | | |
| C2.22 | Students with a disability often abuse the support that they receive. | | | 0.756 | | | | |
| C2.24 | I think that many teachers show too much understanding for the behaviour of students with a disability. | | | 0.677 | | | | |
| C2.33 | Underprivileged parents are often less interested in their children's progress in school than other parents. | | | | 0.669 | | | |
| C2.34 | It is difficult to be sympathetic towards poorer students if they or their parents always have the latest mobile phones or gadgets. | | | | 0.618 | | | |
| C2.35 | Schools that educate many underprivileged students have lower standards. | | | | 0.697 | | | |
| C2.36 | Underprivileged students rarely succeed in difficult subjects because they do not have supportive parents. | | | | 0.590 | | | |
| C2.37 | It is acceptable for teachers to have higher expectations of students from well-off backgrounds. | | | | 0.545 | | | |
| | | | | | | | | |
| C6.01 | The school should provide specific services for students who want to follow religious practices (e.g., prayer room, halal/kosher food). | | | | | 0.508 | | |
| C6.02 | The school must allow students with a migration background to speak a language other than Dutch at school. | | | | | 0.738 | | |
| C6.04 | The school library should have books in the different home languages of the students. | | | | | 0.676 | | |
| C6.05 | It is important to encourage students of immigrant origin to be proud of their ethnic and cultural background. | | | | | 0.513 | | |
| C6.07 | It is time that we give children with a disability a full place in mainstream education. It is their right. | | | | | | 0.765 | |

*(Continued)*

**Table 3.** (Continued)

| To what extent do you agree with . . .? | | Beliefs on different types of diversity | | | | Schools' responsibility for EfA | | |
|---|---|---|---|---|---|---|---|---|
| | | Ethnic minorities | Gender | Disability | SES | Ethnic minorities | Disability | General policy |
| C6.08 | It is sensible to not send students to special educational support too quickly. | | | | | | 0.671 | |
| C6.09 | The school's infrastructure must be made accessible at all levels to people with physical and sensory disabilities (vision, hearing, motor). | | | | | | 0.548 | |
| C6.10 | You cannot expect the teacher to meet the needs of a child with a disability in mainstream education. | | | | | | 0.580 | |
| C6.13 | Teachers must be capable of working with students from different home situations. | | | | | | | 0.608 |
| C6.14 | Students from families with financial problems should be provided with appropriate support (e.g. payment plan, special support fund for students who cannot pay the school fees). | | | | | | | 0.688 |
| C6.16 | A teacher can make a difference for students from all kinds of home environment. | | | | | | | 0.527 |
| C6.21 | As a school, it is important to break the traditional divide between subjects that are stereotypically for boys or for girls. | | | | | | | 0.571 |
| C6.44 | It is every teacher's job to contribute to a school policy that is mindful of diversity. | | | | | | | 0.629 |

## Discussion

While the pursuit of inclusive education is influencing policy agendas of countries across the globe [3], it can be considered a wicked problem as it is characterised by normative plurality, institutional complexity and scientific insecurity [8,10]. This article aims to help grasp the issue by validating the DISCO-instrument, which measures teachers' beliefs and sense of efficacy towards diversity in education. In what follows, we discuss not only the factorial structure arising from the data, but also explore what this unveils on teachers' thinking about diversity in the classroom.

The questionnaire consists of two parts: beliefs and sense of efficacy. Teacher beliefs consist of 36 items in total, and includes beliefs about student diversity and beliefs about the

**Table 4. Descriptive statistics of scale-scores.**

| | Mean | SD | Range | N |
|---|---|---|---|---|
| **Efficacy in** | | | | |
| Noticing diversity | 5,06 | 1,48 | 0–8 | 524 |
| Enabling interactions | 5,88 | 1,23 | 0–8 | 524 |
| Creating learning environments | 5,32 | 1,34 | 0–8 | 524 |
| Collaborating with parents | 5,09 | 1,55 | 0,13–8 | 524 |
| Collaborating with colleagues | 5,89 | 1,56 | 0–8 | 524 |
| **Beliefs on** | | | | |
| Ethnic minority students | 3,78 | 0,98 | 0,5–6 | 520 |
| Students with a disability | 4,20 | 1,00 | 1–6 | 520 |
| The gender binary & sexuality | 4,83 | 0,85 | 1,75–6 | 520 |
| SES students | 3,38 | 1,10 | 0,6–6 | 520 |
| General school policy | 5,00 | 0,70 | 2,40–6 | 520 |
| Policy geared towards ethnicity | 2,93 | 1,21 | 0–6 | 520 |
| Policy geared towards disability | 3,49 | 1,12 | 0–6 | 520 |

responsibility of education to create education for all. The beliefs on student diversity consist of several dimensions, specifically teachers' professional beliefs about students with an ethnic minority background, with a disability, from lower socio-economic backgrounds, and essentialist notions on the gender binary and sexuality. While these dimensions match with typically delineated categories of at-risk students, this was not entirely the case for gender and sexual orientation. Interestingly, even though items on both LGBT students and gender norms were included, analyses show that both pre- and in-service teachers tend to think of these as one overarching construct. This is in line with the theoretical concept of heteronormativity, which highlights the way in which normative ideas on gender expression and sexuality are interwoven [58]. Hence, the validation of the DISCO-instrument lends further support to the construct of heteronormativity, not just from a theoretical perspective but within the minds of teachers as well.

The responsibility-items tap the extent to which teachers believe it is the responsibility of the educational field to create maximal developmental opportunities for diverse students. This includes several dimensions, including one about general school policy, which encompasses both general items (e.g., "It is every teacher's job to contribute to a school policy that is mindful of diversity") as well as items referring to SES- and gender-issues (e.g., "Students from families with financial problems should be provided with appropriate support (e.g. payment plan, special support fund)", "As a school, it is important to break the traditional divide between subjects that are stereotypically for boys or for girls."). Other dimensions encompass items on school policy directed towards specific diversity-groups, with one dimension specifically geared towards students with a migration background, and another geared towards students with a disability. This factorial structure suggests some interesting things about the way teachers think about school policy and the educational field's responsibility on EfA. For instance, issues regarding SES and gender are apparently being regarded as "commonplace enough" to be integrated within general school policy. Conversely, the needs of ethnic minority students and those with a disability are seen as being "too different" and warrant their own specific policies and guidelines. Previous research has shown that Flemish teachers tend to have a lot of questions on how to handle students' multilingualism and religiosity [47]. Furthermore, Flanders historically has a high rate of students segregated into special education when compared to other European nations [59], and has only recently made substantial steps in integrating students with a disability in mainstream education [47]. Hence, it stands to reason that this is contributing to teachers seeing policies for both ethnic minority students and those with a disability as being "special". It would be interesting for future research to explore to what extent teachers in other contexts share a similar view. Hence, we call for a cross-national validation of the DISCO-instrument.

When taking a closer look at how teachers score on belief-scales, some interesting findings emerge. First, teachers tend to score highest on the responsibility of schools to contribute to general school policies for diversity, and have progressive beliefs about students' gender expression and sexuality. The positive stance of Flemish teachers with regard to gender norms and LGBT-rights is in line with previous research [47], and matches the overall progressive stance of Belgium when it comes to LGBT-rights [60]. On the other side of the spectrum, Flemish teachers tend to think least positively about school policies specifically geared toward ethnic minority students and those with a disability, as well as having less positive beliefs about low-SES students. Noteworthy is how teachers combine positive views on general school policies (which include items on providing for low-SES students) with less approving personal beliefs on SES-students. Perhaps this shows that teachers are in support of the uplifting role that education could play and in which strong, general policies have a substantial role, while at the same time acknowledging the day-to-day challenges

they experience when providing education to students from low-SES family backgrounds [11]. Qualitative research would be well-placed to thoroughly investigate and make sense of this ambiguous stance of teachers.

The second part of the questionnaire contains 36 items considering teachers' sense of efficacy, and is organized in five dimensions. The first dimension taps teachers' efficacy in noticing student diversity (f.i., students' background, interests, well-being, . . .), the second the extent to which teachers feel capable to design a classroom which capitalizes on the strengths and needs of their students, and the third measuring the extent to which teachers feel capable to create high-quality interactions among students. The fourth and fifth dimension tap teachers' perceptions of the degree to which they feel capable to collaborate with colleagues and diverse parents respectively.

The analyses on the scale-scores show that teachers tend to feel most efficacious in both collaborating with colleagues and in creating high-quality interactions among students. This is followed by creating stimulating learning environments for diverse students. Note that this dimension includes a lot of items related to providing differentiated instruction, with using a variety of instructional and evaluation methods, and stimulating the interest and engagement of students. As such, this dimension connects to the core of teachers didactical competencies [61,62]. Consequently, the ranking of these dimensions may come as no surprise, with teachers feeling reasonably efficacious in this dimension, while at the same time continuing to see opportunities for further growth and improvement. At the same time, noticing student diversity and collaborating with diverse parents are the lowest rated efficacy-dimensions. Nevertheless, in order to capture students' attention and engagement, theories on differentiated instruction emphasize the importance of taking into account the social environment of students and their individual interests [62]. We could speculate that the relatively low score on noticing diversity reflects the idea among teachers that this is not a must, but merely a nice extra, second to more concrete didactical approaches. Qualitative research with teachers would be well-placed to explore the reasons for teachers' efficacy-assessment on this dimension. Furthermore, the low score on collaborating with diverse parents is in line with previous research in Flanders, indicating that parents' involvement is often only expected or even tolerated under specific circumstances, such as coming to the parent-teacher contact moment or helping out with the school fair [63]. Nevertheless, research has shown that creating strong connections between the school and the home environment is conducive to students' school belonging and achievement [64,65], and could be an important lever in creating quality education for all [11].

Note that the scales on efficacy in collaboration with parents and colleagues originally started out as responsibility belief-scales, which had to be reworked due to ceiling-effects. Interestingly, the data shows that teachers combine the highest efficacy-ratings in collaborating with colleagues with the lowest efficacy-scores in collaborating with parents. This shows both the nuances and complexities in (student-)teachers' assessments of what they *can* and *should* do. That is, our validation process suggests that (student-)teachers are able to recognize that collaboration is something they should do, while simultaneously admitting to low abilities in collaborating with one partner and high scores in collaborating with another.

In conclusion, this article discusses the DISCO-instrument, which measures teachers' beliefs and efficacy for creating quality education for diverse students. In the context of EfA being considered a wicked problem, we suggest that DISCO cannot only help researchers to grasp the state of affairs of a specific school or educational system, but can support policy makers and principals in making evidence-based decisions on the professionalisation needs of their school teams.

## Limitations

While this instrument is able to grasp teachers' beliefs for inclusive education as well as a self-assessment of their skills, it remains to be seen how the DISCO-instrument connects to other aspects of competency such as knowledge and actual classroom behaviour. Furthermore, the validation-process in this study remains largely focused on structural validity, while it has been argued that a full validation-process should encompass a.o. the stability of the instrument, as well as concurrent, discriminant and criterion-related validity [66]. Especially a cross-national comparison could give us some interesting insights into how contexts, with their specific educational realities and policies, shape teachers' thinking and competences on inclusive education. Similarly, it would be interesting to explore to what extent respondents report different beliefs and self-efficacy scores depending on their own intersectional positions. We were unable to explore this as our data did not include demographic information beyond respondents' age, gender and educational background. However, it stands to reason that our sample included few voices from minoritized groups as the Flemish teaching population tends to consist overwhelmingly of white, middle class, cisgender, straight, able-bodied women, with for instance only 3.2% of the teaching population being of ethnic minority descent [67].

Lastly, we suggest that mixed method or qualitative methods might be interesting to further understand the processes underlying teachers' answers to the DISCO-survey and hence their beliefs and self-efficacy assessments regarding inclusive education. For instance, future research, could use think-aloud procedures to gain a deeper insight into how teachers think about specific features of inclusive classrooms, including how they assess the importance of noticing diversity or their ambiguous stance regarding professionals beliefs versus policy aimed at low-SES students.

## Author Contributions

**Conceptualization:** Wendelien Vantieghem, Iris Roose, Karin Goosen, Piet Van Avermaet.

**Data curation:** Wouter Schelfhout.

**Formal analysis:** Wendelien Vantieghem.

**Investigation:** Wendelien Vantieghem, Iris Roose, Piet Van Avermaet.

**Methodology:** Wendelien Vantieghem, Karin Goosen.

**Supervision:** Piet Van Avermaet.

**Writing – original draft:** Wendelien Vantieghem.

**Writing – review & editing:** Wendelien Vantieghem, Iris Roose, Wouter Schelfhout.

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
