## [Decision Letter · Decision Letter 0]

20 Mar 2023

PONE-D-22-31586Education for all in action: Measuring teachers’ competences for inclusive educationPLOS ONE

Dear Dr. Vantieghem,

Thank you for submitting your manuscript to PLOS ONE. After careful consideration, we feel that it has merit but does not fully meet PLOS ONE’s publication criteria as it currently stands. Therefore, we invite you to submit a revised version of the manuscript that addresses the points raised during the review process. The revisions should, however, be very minor. We suggest to especially pay attention to the comments for the first reviewer. This reviewer also added annotations to your original manuscript. Please, e-mail me if you couldn't see these annotations (pieter-paul.verhaeghe@vub.be).

We look forward to receiving your revised manuscript.

Kind regards,

Pieter-Paul Verhaeghe

Academic Editor

PLOS ONE

Journal Requirements:

This research did not receive any specific grant from funding agencies in the public, commercial, or not-for-profit sectors.

Reviewer's Responses to Questions

**Comments to the Author**

1. Is the manuscript technically sound, and do the data support the conclusions?

Reviewer #1: Yes

Reviewer #2: Yes

2. Has the statistical analysis been performed appropriately and rigorously? 

Reviewer #1: Yes

Reviewer #2: Yes

3. Have the authors made all data underlying the findings in their manuscript fully available?

Reviewer #1: Yes

Reviewer #2: Yes

4. Is the manuscript presented in an intelligible fashion and written in standard English?

Reviewer #1: No

Reviewer #2: Yes

5. Review Comments to the Author

Reviewer #1: Reviewer comments on PONE-D-22-31586: Education for all in action: Measuring teachers’ competences for inclusive education

Thank you for the opportunity to review this interesting manuscript, and do accept my apologies for being a few days late with the review.

This article makes a valuable contribution to the field of inclusive education, specifically within a European (Flemish) context and with the aim of developing a quantitative instrument to add to the largely qualitative body of research in this – as the authors rightly describe it – “wicked” area. The article is on the whole well-written, the literature review flows cleanly into the description of the study, and the discussion and conclusions complement the data well, so that the reader can clearly understand and interpret the findings.

I am in no doubt that this manuscript can be brought up to the standard required for publication. I do, however, have a number of suggestions as to how to further strengthen the piece, some very minor and some that will require more thought and action. I have annotated the attached PDF and hope that the comments will largely speak for themselves. I do wish to place additional emphasis on the following points (which are also in the document), pertaining to the sensitive nature of this branch of education:

1. My comment on p.10: I have no particular problems with the scales based on the items cited (but do see my small comments on some of the items in the tables at the end), and can see how the choice of focus on self-efficacy and beliefs follows on from the literature reviewed in the previous section. What I am missing, though, is a section in which you explain from which standpoint/perspective the individual items were formulated: in other words, how are you defining 'inclusive education' or EfA? Do you take a social justice standpoint? Are you drawing on principles from a particular model or author, or from a combination of different sources? I notice, for example, that the items in the tables at the end do not relate to areas such as anti-racist education, content integration or critical literacy, which are present in many of the well-used models of inclusive education. If the survey is measuring self-efficacy in inclusive education (or even, competence), it would be helpful to know what you are positioning as effective practice or, as you mention earlier, "the educational preconditions necessary for students’ maximal learning" as part of the inclusive context.

2. In the questionnaire items in the tables, I have made a number of comments regarding what look like misconceptions in the wording. I realise that the instrument can no longer be altered, but on some occasions I wonder if the problem is with the English translation rather than the item itself. In that case, they can easily be repaired. I urge the authors to look at this carefully, and if the translation cannot be repaired, to consider acknowledging these weak points somewhere along the way (or else defending the wording, if there are arguments in its favour). A few other issues with wording (e.g. using LGBT and gender interchangeably) come up elsewhere as well. I strongly advise that the authors take care to iron these out.

3. Some aspects of the ‘story’ of the questionnaire development process were unclear. I have tried to point these out from the perspective of the reader, sometimes at the risk of sounding rather stupid myself!

4. The different roles of the pre- and in-service teachers were not clear to me. This came up a few times.

5. I found a number of inconsistencies and contradictions in the text, which I have flagged in the comments.

For further comments and specific examples, see the attached copy of the manuscript.

I wish the authors the very best of luck in revising this paper and look forward to seeing either the next version when it is ready.

Reviewer #2: The submission makes an important contribution to the field of teaching in diverse classrooms. The study is conducted according to scientific rigorous standards and the paper is written in a clear, concise and yet inviting style. I would like to make a few minor suggestions:

*I miss a reference to an often used teacher self-efficacy distinction: In most research TSE includes three components, namely efficacy in instructional practices, classroom management and student engagement (see e.g., Zee and Koomen, 2016). There may be good reasons to choose a different distinction, but the authors should refer to the more classical distinction and explain their choice.

*Furthermore, the authors should also connect to the recent research on diversity related self-efficacy: see Romijn et al. (2020) (who provided evidence for a two-dimensional nature of TSE, with a distinction between general TSE and diversity-related TSE) and Siwatu (2011) who describe a specific measure for the concept of culturally responsive teaching self-efficacy.

*In the introduction, the authors suggest they are going to develop an instrument that will measure skills or capabilities. However, in the remainder of the paper it becomes clear that DISCO measures self-efficacy instead. This should be made clear from the start.

*I miss information (in a table) on the initial 125 items, the deletion of items in each step, and the final number of items. Both in terms of numbers, as well as in terms of content. By content I mean the reference to underlying theory. The initial items were designed on the basis of theory, and experts expected these items to be a valid reflection of the underlying constructs. However, the psychometric analyses showed the experts were ‘wrong’ about many items. I would like to suggest the authors reflect on this process. Did the authors deleted items solely based on psychometric properties? Or did theory also play a role in their decisions (as it should).

*Aside from C2.01, C2.14 and C2.27, all beliefs are negative beliefs. Why did the authors choose this approach? And why not the Hachfeld approach of two positively framed dimensions (egalitarianism and multiculturalism). Surely this has impacted the results. I think the authors should explain their choice, and the discussion needs a critical reflection on this choice and its consequences.

*What was the response rate in the first data collection?

*In what cases were respondents deleted from the data: there were quite many (attrition rates of 33% and 15%). On p. 13 the authors write that means were computed on scales with a max of 25% missing values. This suggests that respondents with missing values are still in the database. So how many missing values per respondent were allowed?

6. PLOS authors have the option to publish the peer review history of their article (what does this mean?). If published, this will include your full peer review and any attached files.

Reviewer #1: No

Reviewer #2: No

---

## [Author Response · Author response to Decision Letter 0]

15 Jun 2023

Response to reviewers

Note to the Editor

We thank the editor and reviewers for their interesting feedback and the chance to re-work the paper. We feel that thanks to the suggestions of the reviewers, the manuscript has improved and some issues concerning clarity and steps taken throughout the validation process have become more clear. Below, we explain in detail how we engaged with each point the reviewers made. Note that in response to a question of both reviewers, a rather substantial table has been added that lists all items originally included in the DISCO-instrument (see Table 1). We leave it up to the discretion of the editor and the journal whether this table works better included within the body of the text or added as an appendix to the manuscript. Additionally, thanks to the suggestions made by the reviewers, the reference list has been updated with some additional citations as well. This can be seen throughout the manuscript and in the bibliography, but for clarity, we also provide a list of additionally included sources below:

• Billiet, J., & Waege, H. (2003). Een samenleving onderzocht: Methoden van sociaal-wetenschappelijk onderzoek. De Boeck Hoger. 

• Commissie_Diversiteit. (2020). Diversiteit binnen het onderwijzend personeel. Brussels: SERV Retrieved from https://publicaties.vlaanderen.be/view-file/38452

• Davis, H. A. (2003). Conceptualizing the role and influence of student-teacher relationships on children's social and cognitive development. Educational psychologist, 38(4), 207-234. 

• Gheyssens, E., Consuegra, E., Vanslambrouck, S., Engels, N., & Struyven, K. (2020). Differentiated instruction in practice: do teachers walk the talk? Pedagogische Studiën, 97(3), 163-186. 

• Hoy, A. W., Hoy, W. K., & Davis, H. A. (2009). Teachers’ self-efficacy beliefs. In K. R. Wentzel & A. Wigfield (Eds.), Handbook of motivation at school (pp. 641-668). Routledge. 

• Newsom, J. (2023). Missing Data and Missing Data Estimation in SEM. In J. Newsom (Ed.), Structural Equation Modeling. Portland, USA: Portland State University.

• Ponet, B., Tack, H., Vantieghem, W., Van Avermaet, P., & Vanderlinde, R. (2021). Hoe lerarenopleiders omgaan met diversiteit: naar een conceptueel raamwerk. Tijdschrift voor Lerarenopleiders, 42(4), 45-60. 

• Romijn, B. R., Slot, P. L., Leseman, P. P., & Pagani, V. (2020). Teachers’ self-efficacy and intercultural classroom practices in diverse classroom contexts: A cross-national comparison. International Journal of Intercultural Relations, 79, 58-70. 

• Sharma, U., Loreman, T., & Forlin, C. (2012). Measuring teacher efficacy to implement inclusive practices. Journal of Research in Special Educational Needs, 12(1), 12-21. 

• Siwatu, K. O. (2007). Preservice teachers’ culturally responsive teaching self-efficacy and outcome expectancy beliefs. Teaching and Teacher Education, 23(7), 1086-1101. 

• Van Avermaet, P., Van Houtte, M., & Van den Branden, K. (2011). Promoting equity and excellence in education. An overview. In K. Van den Branden, P. Van Avermaet, & M. Van Houtte (Eds.), Equity and excellence in education. Towards maximal learning opportunities for all students (pp. 1-20). Routledge. 

• Zee, M., & Koomen, H. M. (2016). Teacher self-efficacy and its effects on classroom processes, student academic adjustment, and teacher well-being: A synthesis of 40 years of research. Review of Educational Research, 86(4), 981-1015.

Reviewer 1

This article makes a valuable contribution to the field of inclusive education, specifically within a European (Flemish) context and with the aim of developing a quantitative instrument to add to the largely qualitative body of research in this – as the authors rightly describe it – “wicked” area. The article is on the whole well-written, the literature review flows cleanly into the description of the study, and the discussion and conclusions complement the data well, so that the reader can clearly understand and interpret the findings.

I am in no doubt that this manuscript can be brought up to the standard required for publication. I do, however, have a number of suggestions as to how to further strengthen the piece, some very minor and some that will require more thought and action. I have annotated the attached PDF and hope that the comments will largely speak for themselves. I do wish to place additional emphasis on the following points (which are also in the document), pertaining to the sensitive nature of this branch of education:

Comment 1

My comment on p.10: I have no particular problems with the scales based on the items cited (but do see my small comments on some of the items in the tables at the end), and can see how the choice of focus on self-efficacy and beliefs follows on from the literature reviewed in the previous section. What I am missing, though, is a section in which you explain from which standpoint/perspective the individual items were formulated: in other words, how are you defining 'inclusive education' or EfA? Do you take a social justice standpoint? Are you drawing on principles from a particular model or author, or from a combination of different sources? I notice, for example, that the items in the tables at the end do not relate to areas such as anti-racist education, content integration or critical literacy, which are present in many of the well-used models of inclusive education. If the survey is measuring self-efficacy in inclusive education (or even, competence), it would be helpful to know what you are positioning as effective practice or, as you mention earlier, "the educational preconditions necessary for students’ maximal learning" as part of the inclusive context.

Response 1

We thank the reviewer for this interesting question to expand upon the underlying theoretical framework to inclusive education that this study adheres to. We follow the perspective on inclusive education that is commonly called Education for All (EfA), which explicitly stems from a social justice perspective on education (and society as a whole) (Ainscow & Miles, 2008; Opertti et al., 2014). 

We are of the view that inclusive education and diversity are two sides of the same coin. Within this view, diversity is understood as the myriad ways in which people differ from each other (Van Avermaet & Sierens, 2010). Nevertheless, diversity does not happen in a social vacuum but is present in societies with existing power relations and structures that favour people with certain identity markers over others (Crenshaw, 2003; McIntosh, 1990), resulting in systematic inequalities in society and education. Consequently, responsiveness to diversity in education is twofold: it is taking into account differences between people in order to create qualitative learning environments for everyone, as well as responding to discriminatory injustices that exist in society and (in)directly impact education in order to create a more equitable world. (Ponet et al., 2021).

This view on diversity in education within a social justice perspective is currently aptly captured within the movement for inclusive education and more specifically Education for All (EfA). That is, the evolution of the concept of inclusive education shows a progressing understanding of how unequal educational opportunities for specific groups of students are linked together and to the general functioning of educational systems, with a simultaneous comprehension of the importance of an individual rights-based perspective to avoid issues of categorization and essentialization (Opertti et al., 2014). In order to achieve this transformation, it is important to consider inclusion as a continuous process of improvement, in which all (educational) stakeholders have a responsibility and role to play. As such, inclusive education is something which each teacher can and should strive towards, not just something that some teachers can discuss with their student during specific classes focused on topics of diversity. (Note that this reductionist view on diversity-sensitive education is something oft-encountered among educational stakeholders, who delegate the responsibility for inclusion or diversity-sensitiveness to teachers with specific roles, such as school counsellors, or specific courses such as history or language (Groenez et al., 2018; Ponet et al., 2021)). Similarly, the EfA-paradigm has identified changing stakeholders’ mindset or beliefs as being of the utmost importance, so issues are seen as being located within the system and not within specific (groups of) children (i.e., a deficit perspective). In order to differentiate between this view and older conceptualisations of inclusion, the concept of EfA was launched. (we like to refer to the insightful description of Opertti et al., 2014 for a detailed discussion of this evolution of the concept of inclusion into EfA). 

Hence, this article subscribes to this view on inclusion and diversity put forward within EfA, which has been inspired and expaned upon by several authors including Opertti et al. (2014), Dyson et al. (2002), Ainscow and Miles (2008), Messiou (2017), Van Avermaet et al. (2011) and so on, and which has inspired the conceptualisation of the study and operationalisation of the DISCO-instrument. As the reviewer aptly notes, this view impacts not only the theoretical underpinnings of the study but has an important influence on the practices considered as contributing to inclusion and consequently items that were formulated for the DISCO-instrument. A detailed discussion of the paradigm of EfA and the evolution of the concept of inclusive education is beyond the scope of this article. Nevertheless, we concur with the reviewer that it would be insightful to the reader to get a better grasp of these theoretical underpinnings and how they relate to the construction of the items. Consequently, we have included a more detailed discussion of EfA and what this framework means for the conceptualization and choices made within the study throughout the manuscript. This can be seen in the excerpts below and on page 2 in the introduction, on page 7 when discussing teachers’ self-efficacy, and on page 11 in the methodology when discussing item creation.

While inclusion was initially focused on groups traditionally occupying a marginalized position in education (including students with disabilities, from ethnic minority descent or disadvantaged families), inclusive education is currently conceptualised as a call to transform educational systems at large to reach all students regardless of their background (for a detailed discussion of the evolution of the concept of inclusive education, see Opertti et al., 2014). This perspective on inclusion explicitly starts from a social justice standpoint and is known as Education for All (EfA) (Ainscow & Miles, 2008). Important within this view is the recognition of the educational fields’ responsibility to take into account differences between learners in order to create qualitative learning environments for everyone, while maintaining a clear understanding of how systemic inequalities in education disproportionally affect specific (groups of) students (Ponet et al., 2021).

In order to realise the maximal learning and developmental outcomes suggested within EfA, inclusion cannot be relegated to the side-lines as being something that some teachers sometimes do within the context of specific courses or lessons, or something that only expert teachers do after having achieved mastery in “general” teaching tasks aimed at “regular” students (Van Avermaet et al., 2011). Rather, it becomes an inclusive culture that all teachers adhere to in the classroom (for a more detailed discussion of inclusive school cultures, see Ainscow et al., 2006), and which requires a constant effort of teachers to continuously monitor, evaluate and adapt their teaching methods to best suit the needs of the students present (Giangreco et al., 1995). In such an endeavour, self-efficacy could constitute a powerful resource, as self-efficacy beliefs have proven to be powerful predictors of teachers’ motivation and actual performance (Klassen et al., 2011) with a high sense of self-efficacy being especially powerful in the face of adversity, in order to persevere when faced with difficulties or failure (Bandura, 1997).

First, a pool of items was created by three researchers with expert knowledge on diversity in education. In order to maintain the goals of the DISCO-instrument and reflect the central tenets of EfA, the following criteria were observed during this creation phase,: (1) to mitigate the fragmentation of the EfA field and employ a broad definition of diversity, items had to either be applicable to all children or refer to specific diversity groups in a balanced way; (2) items needed to explicitly measure a single practice or perception; and (3) items needed to be relevant across multiple teaching contexts; this implied that items had to be recognisable for teachers across different teaching levels, including primary and secondary; that items had to refer to actions that could be performed by all teachers, and not just those teaching specific courses or students; and had to refer to actions pertaining to day-to-day teaching activities in order to reflect the continuous nature of working on inclusion.

Comment 2

In the questionnaire items in the tables, I have made a number of comments regarding what look like misconceptions in the wording. I realise that the instrument can no longer be altered, but on some occasions I wonder if the problem is with the English translation rather than the item itself. In that case, they can easily be repaired. I urge the authors to look at this carefully, and if the translation cannot be repaired, to consider acknowledging these weak points somewhere along the way (or else defending the wording, if there are arguments in its favour). A few other issues with wording (e.g. using LGBT and gender interchangeably) come up elsewhere as well. I strongly advise that the authors take care to iron these out.

Response 2

We thank the reviewer profoundly for noting these issues. Indeed, the reviewer was quite correct in raising the idea that these issues might stem from translation-issues from Dutch to English, which was actually the case for all flagged items. We asked an English native-speaker to create the translation of the survey items, however, the translator took some liberties here and there and in some cases certain nuances got lost in translation. One of these, for instance, was the use in some items of the popular Dutch acronym “holebi”, which stands for gay, lesbian and bisexual people, which the translator adapted to the LGBT-acronym for the English items (and hence, adding the T for transgender in the process even though this was not part of the original item).

Similarly, we thank the reviewer for noting the way we referred to gender and LGBT throughout the manuscript and how this could lead to confusion with the reader. Consequently, we have taken care to be specific in our wording throughout the manuscript when either discussing issues relating to sexual orientation or gender. Where these issues had to be discussed in tandem (i.e., due to items on LGBT-students and gender norms loading together onto one dimension), we referred to this as “essentialist notions on the gender binary and sexuality”, which we felt managed to capture the content of the items well as they tapped traditional views on heterosexuality, gender norms, identity and expression, and which allowed us to wait until the discussion-section to delve deeper into the theoretical concept of heteronormativity. 

Comment 3

Some aspects of the ‘story’ of the questionnaire development process were unclear. I have tried to point these out from the perspective of the reader, sometimes at the risk of sounding rather stupid myself!

Response 3

We thank the reviewer for diligently pointing out all these issues which we have resolved throughout the text. For instance, we have taken care to ensure that the discussion of beliefs and self-efficacy now happens in a consistent fashion throughout the manuscript by always first discussing beliefs before discussing self-efficacy throughout the literature review, methodology, results and discussion. Where we felt a bit of additional explanation was required in order to substantiate the changes or choices made, we discuss these below (specifically in response to comment 5).

Comment 4

The different roles of the pre- and in-service teachers were not clear to me. This came up a few times.

Response 4

We thank the reviewer for noting this issue with clarity. We have made the set-up with two different samples, being pre-service teachers on the one hand and in-service teacher on the other, more clear throughout the text. A full explanation is given at the start of the methodology, and is repeated in the plan of analysis to remind the reader of the used approach. This can be seen on pages 9 through 14 in the manuscript and in the excerpts below. We hope that these adaptations have made the used approach clearer to the reviewer and future readers.

Sample & respondents

“In order to allow for a process of trial and improvement as well as cross-validation, data was collected in two phases within Flanders (Belgium). In the first phase, data was collected among pre-service teachers. Based on the exploratory factor analyses on this data, the survey was optimised and then tested again in the second phase. During this second phase, data from a new sample consisting of in-service teachers was analysed using confirmatory factor analyses (for more information, see Plan of analysis).”

Plan of analysis

To validate the factor structure underlying the DISCO-instrument, we combined exploratory (EFA) and confirmatory factor analyses (CFA) in a stepwise approach. In the first step, using the data from pre-service teachers, exploratory factor analysis using SPSS 25 was chosen to identify a (set of) latent constructs…

…

For the second step, the survey was adapted according to the insights from the first step and then presented to a new sample consisting of in-service teachers. The initial factor solution from step 1 was checked with data from these in-service teachers using confirmatory factor analyses in Mplus. 

…

Finally, based on these results, the scale scores for in-service teachers were obtained by computing a mean for each respondent with a maximum of 25% missing values on the scale-items.

Comment 5

I found a number of inconsistencies and contradictions in the text, which I have flagged in the comments.

Response 5:

We thank the reviewer profoundly for consistently indicating these issues. We have adapted all these issues as suggested by the reviewer. Below, we discuss some of these in-text adaptations where changes constituted bigger adaptations or where we felt a bit of additional explanation was warranted to explain the underlying choices we made.

Overview of in-text suggestions by the reviewer and author’s responses:

- Pg 5 “However, teachers tend to report more negative attitudes towards inclusive education than either pupils or parents (Van Mieghem et al., 2020), …”

big statement based on only one previous study: it would be wise to use some hedging here

o Response 6: The study by Van Mieghem and colleagues concerns a systematic search and meta review of studies on inclusive education from across the globe from 1980 to 2015. In this sense, this statement is not so much based on one study, but rather a representation of a trend observed in studies on inclusive education. We thank the reviewer for noting the importance of clarifying this, and have adapted the sentence to reflect that: “However, a meta-review of research in-between 1980 and 2015 shows that teachers tend to report more negative attitudes towards inclusive education than either pupils or parents (Van Mieghem et al., 2020), …”

- Pg 8: “Hence, the goal of this article is to establish a valid and reliable instrument to measure teachers’ beliefs and self-efficacy with regards to inclusive education.” � Originally you said it was about measuring their competences; here you say it is about measuring beliefs and self-efficacy. I actually found 'competences' somewhat problematic as a self-report instrument will only tell us how the teacher views their own competence, not how they behave in practice. I actually think that the way it is portrayed in this sentence is clearer and more accurate: it is about beliefs and self-efficacy (and I see nothing wrong with that).

o Response 7: we thank the reviewer for this acknowledgement and agree wholeheartedly. In order to make these considerations clear to all readers, the manuscript includes the section on teachers’ competences in inclusive education, where we discuss several operationalisations of the concept of “competence”. There, we clarify that this manuscript subscribes to the tradition which employs a broader definition of competence, pointing to the process resulting in observable behavior, in which dispositions and situation-specific skills are of central importance, and hence, in which beliefs and self-efficacy are of central importance as building blocks of competence.

- Pg 8: “I don't suppose you have data on the numbers of participants who belonged to the minoritised groups focused on by the instrument? Given the nature of the study, that could well play a role. If you don't have this data, you can't do anything about it now, although it might be worth mentioning as a limitation of the study.”

o Response 8: unfortunately, the reviewer is correct in thinking that we do not have this kind of demographic information on our respondents. Following the suggestion of the reviewer, we have included this as a limitation in the discussion-section, as can be seen on page 27 in the manuscript and in the excerpt below.

“Similarly, it would be interesting to explore to what extent respondents report different beliefs and self-efficacy scores depending on their own intersectional positions. We were unable to explore this as our data did not include demographic information beyond respondents’ age, gender and educational background. However, it stands to reason that our sample included few voices from minoritized groups as the Flemish teaching population tends to consist overwhelmingly of white, middle class, cisgender, straight, able-bodied women, with for instance only 3.2% of the teaching population being of ethnic minority descent (Commissie_Diversiteit, 2020).”

- Pg 10: “Were there only negative statements or also positive ones? Or stereotypes that are not obviously positive or negative? It would be helpful to explain the type of statements a bit better, including some discussion of why this was the chosen approach.”

o Response 9: We thank the reviewer for this comment and like to refer them to our response to comment 5 of Reviewer 2 for an answer.

- Pg 11: “I see that this section also contains items referring to the school as an organisation, not just the role of the individual teacher. In the literature review, it sounded as though a distinction would be drawn between these two levels, but here they seem to be grouped together. Which is it and why?”

o Response 10: 

The reviewer is quite correct in noting that the DISCO-instrument set forth to measure beliefs on the educational fields’ responsibility to provide EfA. As DISCO is a survey-instrument, these responsibility beliefs are measured on the level of individual teachers’ views on these matter. However, as the reviewer notes correctly, the items underlying this factor assessed both actions on the level of what individual teachers should do as on the level of what schools as a whole can do (for instance with regards to policy). Following the tenets of EfA (as discussed in our response to comment 1), it was important to incorporate both levels in the responsibility beliefs as inclusion can only be reached when individual teachers consider this a task that each and every one of them can work on continuously in the classroom, while at the same time acknowledging the fact that systematic inequalities that are present in the context of education necessarily need to be tackled on the level of the institutions as a whole as well (i.e., the transformation of education systems). In order to make this more clear, we have adapted the wording throughout the beliefs-section on page 5 of the manuscript and in the methodology-section on page 12 in the mauscript. Furthermore, while both levels were present in the items included in the survey, we hypothesized that actions would be clustered within teachers’ minds based on the specific diversity groups these related to rather than on the specific agent undertaking an action (i.e., either something teachers can do versus something a school as a whole does), which spurred the way this is currently discussed throughout the methodology section. Interestingly, the factor analyses pointed out that the truth lies somewhere in the middle, with on the one hand having a factor relating specifically to the broad role a teacher can take on (which was not retained for the second step due to low cronbach’s alpha), while other factors showing a mix of items with both teachers and schools undertaking an action. At the same time, these are also not completely clustered according to specific diversity groups to which actions of schools’ and individual teachers are aimed, but rather consist of one factor tapping policies from a broad view on diversity and two geared specifically towards specific diversity groups.

- Pg 13: “I feel ready at this point for an overview of what the factors were and which items they contained. Otherwise the procedures have little meaning.”

o Response 11: We thank the reviewer for this comment and like to refer them to our response to comment 4 of Reviewer 2 for an answer, as well as to table 1 in the manuscript which gives an overview of the factors and items underlying the DISCO-instrument, and how changes were made throughout the research steps.

Reviewer 2

Comment 1

The submission makes an important contribution to the field of teaching in diverse classrooms. The study is conducted according to scientific rigorous standards and the paper is written in a clear, concise and yet inviting style. I would like to make a few minor suggestions:

*I miss a reference to an often used teacher self-efficacy distinction: In most research TSE includes three components, namely efficacy in instructional practices, classroom management and student engagement (see e.g., Zee and Koomen, 2016). There may be good reasons to choose a different distinction, but the authors should refer to the more classical distinction and explain their choice.

Response 1

We thank the reviewer for this interesting question. Indeed, we have not used the oft employed distinction in teacher self-efficacy research for the three subcomponents of instructional practices, classroom organisation and student engagement. This choice was made for several reasons. First, the final version of DISCO includes 5 self-efficacy related factors, of which two (being collaboration with diverse parents and colleagues) clearly transcend the classroom-level that is the focus in the traditional distinction of instructional practices, classroom management and student engagement. Secondly, in the view of “education for all” that this article subscribes to, inclusive education is operationalized as creating learning environments in which all students can reach their maximal learning & developmental outcomes. As often noted by many researchers, this does not necessarily require new teaching practices, but rather an awareness of diversity and goal-orientation towards equitable learning opportunities in what and why teachers do what they do (Gheyssens et al., 2020). Besides this awareness & goal-orientation on diversity & inclusion, an important pre-condition is positive interactions and a safe classroom space for (especially at-risk) students, both with regards to motivation & well-being as with regards to learning. Consequently, and as can be noticed when considering the items included in each scale, the self-efficacy related scales in DISCO tend to show a blend of items pertaining to instructional, organizational and emotional factors in order to really capture how and why teachers engage in inclusive practices. For instance, the scale on stimulating learning environments for diverse students includes a range of items relating to using a variety of instructional methods, as well as stimulating the interest and engagement of students. Similarly, the scale on enabling high-quality student-interactions contains a mix of items of which some are more solely related to emotional support, whereas others can be situated in the domain of cooperative learning, which could be considered more closely tied to instructional practices.

Hence, we felt that the DISCO-scales would not be able to truly capture the EfA-vision on inclusion were we to maintain the distinction in-between instruction, management and student engagement. 

Nevertheless, we recognize that it would be interesting for the reader to know the reasoning behind this choice and have added a sentence in the article on this, as can be seen in the excerpt below and on page 8 in the manuscript.

Note that by delineating these aspects, this paper does not explicitly follow the oft-used distinction in teachers’ self-efficacy research in-between instructional strategies, classroom organisation and student engagement (Hoy et al., 2009; Zee & Koomen, 2016). This choice was fuelled by researchers often noting that EfA does not necessarily require new teaching practices, but rather an awareness of diversity and a goal-orientation towards equitable learning opportunities in what teachers do (Gheyssens et al., 2020; Vantieghem & Van de Putte, 2019). For instance, within the aspect of positive classroom interactions can cooperative learning be linked to both student engagement and instructional strategies (Davis, 2003). Hence, we felt that an instrument designed to grasp teachers’ efficacy for inclusive education would be better served by recognizing this complex intermingling of goals and methods rather than a strict distinction in-between instructional strategies, classroom organisation and student engagement.

Comment 2

Furthermore, the authors should also connect to the recent research on diversity related self-efficacy: see Romijn et al. (2020) (who provided evidence for a two-dimensional nature of TSE, with a distinction between general TSE and diversity-related TSE) and Siwatu (2011) who describe a specific measure for the concept of culturally responsive teaching self-efficacy.

Response 2

We thank the reviewer for pointing out these interesting articles. We concur with the reviewer that these studies provide interesting insights, for instance with regards to the connection between general self-efficacy and diversity-specific teaching efficacy. Especially interesting in this respect is the difference between the papers, where Romijn et al. show how these aspects can function independently from each other in a two-dimensional factorial structure; while Siwatu worked with a one-dimensional structure by including items relating to both general & diversity-specific efficacy, and conceptualised general teaching tasks as reflecting the “easy side of the continuum” and culturally sensitive practices reflecting the “difficult side of the continuum” (Siwatu et al., 2008, p. 1089). The current article does not employ such a distinction between general and diversity-specific teaching self-efficacy (see also our response to comment 1 above), as the EfA paradigm explicitly wishes to transform educational systems at large in order to be diversity-sensitive (Ainscow & Miles, 2008) (for more information on the EfA-paradigm the manuscript subscribes to, we also like to refer the reviewer to our response to comment 1 of Reviewer 1). Hence, EfA explicitly overturns the idea of a distinction between so-called “general” and “diversity-specific” practices by stating that quality education cannot be anything else but sensitive towards the diversity that is present in each and every classroom, echoing notions put forward by Van Avermaet and colleagues on how “diversity is the norm” and how diversity should be situated in the core of education, not relegated to the side-lines or treated as another add-on (Van Avermaet & Sierens, 2010). To make this standpoint more clear (as well as in response to comment 1 by Reviewer 1) we have included a more substantial discussion of the EfA paradigm’s views on diversity in education in the introduction-section and the section on teachers’ self-efficacy, and how this reverberates through the operationalisation of the items in the scales, as discussed in the methodology section.

Secondly, while the studies of Romijn et al. and Siwatu present interesting results from the development of scales specifically designed to capture teachers’ self-efficacy with regards to (ethnic) diversity, the goal of this study was to develop scales that engage with a broad understanding of diversity rather than focusing on one specific subgroup, and hence, help mitigate the fragmentation plaguing research on EfA. In the section on teacher beliefs, we had already included a short discussion of this and have now done the same regarding self-efficacy scales which can be seen on page 7 in the manuscript and in the excerpt below. 

Importantly, research has shown that self-efficacy is not unidimensional, but tends to be context- and task-specific (Huang, 2013). Consequently, it is important to consider teachers’ self-efficacy with regards to several necessary tasks to create inclusive classroom environments rather than one overall assessment. At the same time, it remains important to employ a broad definition of diversity in order to truly capture EfA and the intersectionally diverse nature of classroom contexts nowadays, rather than narrowing the focus to a specific diversity group (note that such teaching self-efficacy scales do exist, such as the ones focused on ethnic diversity developed by Siwatu (2007) or Romijn et al. (2020), or focused on students with a disability by Sharma et al. (2012)). Hence, in this research, teachers’ self-efficacy beliefs with respect to several central aspects of realizing inclusive learning environments will be assessed.

comment 3

In the introduction, the authors suggest they are going to develop an instrument that will measure skills or capabilities. However, in the remainder of the paper it becomes clear that DISCO measures self-efficacy instead. This should be made clear from the start.

Response 3

We thank the reviewer for this comment and like to refer to response 7 to Reviewer 1 on page 6 of this document for an answer. We hope that by having this explanation on the operationalization of competence into beliefs and efficacy within the first two pages of the manuscript as well as shortly explained within the abstract will make this issue clear enough to the reader.

comment 4

I miss information (in a table) on the initial 125 items, the deletion of items in each step, and the final number of items. Both in terms of numbers, as well as in terms of content. By content I mean the reference to underlying theory. The initial items were designed on the basis of theory, and experts expected these items to be a valid reflection of the underlying constructs. However, the psychometric analyses showed the experts were ‘wrong’ about many items. I would like to suggest the authors reflect on this process. Did the authors deleted items solely based on psychometric properties? Or did theory also play a role in their decisions (as it should).

Response 4

Both reviewers noted the need for an additional table giving an overview of all original items included in the DISCO-instrument. Consequently, we have included Table 1 in the manuscript which gives an overview of the factors and items underlying the DISCO-instrument, which changes were made throughout the research steps and why these changes were made. As the reviewer correctly notes, decisions to retain, add or drop items were made for both psychometric and theoretical reasons. For instance, when psychometric indicators signalled the possibility to choose between several items in order to resolve a psychometric issue (f.i. when confronted with high (partial) correlations between two items), the decision on which item was either dropped or retained was fuelled by theoretical considerations such as capturing a more central theoretical aspect, being theoretically more closely aligned with the content of the dimension being captured, or being more in line with central tenets of the EfA framework. This underlying reasoning has been made more explicit in the manuscript text, as can be seen on page 14 in methodology and in the excerpt below. Note that we have also included more information on the creation of the initial pool of items and how these were theoretically inspired by the EfA paradigm, as can be seen on page 11 in the manuscript. 

Items were eliminated if they had inadequate (partial) correlations or high cross-loadings (>|.40|), if their variance was not well explained as suggested by low communalities, or if the loading on their designated factor was weak (<|.35|). When a choice for being dropped or retained had to be made between items, the theoretical interpretability and alignment with EfA tenets spurred the decision besides psychometric considerations. 

…

Where necessary, the model was optimized based on suggestions of the modification indices, standardized factor loadings, R-square estimates of the items, and theoretical interpretability of the factor.

comment 5

Aside from C2.01, C2.14 and C2.27, all beliefs are negative beliefs. Why did the authors choose this approach? And why not the Hachfeld approach of two positively framed dimensions (egalitarianism and multiculturalism). Surely this has impacted the results. I think the authors should explain their choice, and the discussion needs a critical reflection on this choice and its consequences.

Response 5

The reviewer is correct in noting that many items in the beliefs-section of the questionnaire are negatively worded. Indeed, this differs from other scales, of which some consist of solely positively worded items (as is the case in the Hachfeld and colleagues approach regarding scales on multiculturalism and egalitarianism, which matched their desire to differentiate between two different conceptualizations underlying favourable beliefs towards ethnic diversity), while others use a mix of positive and negative worded items (e.g., the professional beliefs about diversity by Pohan and Aguilar (2001)).

We consciously opted for The DISCO-instrument to consist of a majority of negatively worded items to combat several possible issues in the validation of the scales, including ceiling effects and social desirability. As we wanted to create a survey that was able to capture the variability within the teacher population, it was important to make it possible for respondents to go against the grain and give an opinion that might be considered more “controversial”. In the context of (Flemish) education, outright anti-diversity or anti-inclusion ideas can be considered more controversial, as teachers as a population tend to be considered and have been shown to be on average rather socially progressive regarding matters of equity and diversity (Groenez et al., 2018). Furthermore, inclusive teaching has become an obligatory professional competence for teachers in many countries, including Flanders (Burns & Shadoina-Gersing, 2010) (which is discussed in the article on page 3). Given the fact that the DISCO-instrument was and is framed towards respondents as a tool to measure their views on diversity in education, we can expect contextual carry-over effects in which an activation of professional norms and obligations becomes more likely (Billiet & Waege, 2003), and hence in which teachers might feel pressured to show their support of policies they are expected to uphold. Adding to this effect is the so-called acquiesce bias, a tendency of respondents to concur with what they suppose is the opinion of both the majority of their peers as well as the researcher (Billiet & Waege, 2003). Because of this, we felt that it was especially important to safeguard the possibility for respondents to give “controversial” opinions by providing them with a majority of items advocating for critical or negative views on inclusion or student diversity, normalizing the idea within teachers’ mind that it is okay to agree with such items.

Note that we still strived towards avoiding a so-called response set, in which respondents automatically tick (dis)agree on all items as they are all oriented in the same direction, by including positively worded items alongside the majority of negatively-worded items. We specifically ensured that each possible sub-scale had at least one positively worded item. However, throughout the validation process, some of these items did not make the final cut for various reasons (including ceiling effects, confirming our idea on the need for negatively worded items). In response to comment four, we have included a table consisting of all original DISCO-items and the reasons for their deletion, which helps to make this process more clear. Furthermore, in order to make the considerations underlying the choice for a majority of negative items explicit to the reader, we have included a note on this in the methodology section, which can be seen on page 12 and in the excerpt below.

Given that inclusive teaching has become an obligatory professional competence, and hence a norm in education, this increased the chance of social desirability bias and ceiling effects in respondents’ answers (Billiet & Waege, 2003). To counteract this, we provided respondents with a large range of items that were critical of inclusion or student diversity, and hence, reverse-coded.

comment 6

What was the response rate in the first data collection?

Response 6

We thank the reviewer for noting this oversight. Consequently, we have added a sentence which adds the participation rate of the teacher education institutions in Step 1 of the research, so that the participation rate in both steps of the research are reported. This can be seen in the manuscript on page 10 and in the excerpt below.

In the first step, data was collected among pre-service teachers in the fall of 2017 in a sample of teacher education institutions. All teacher education institutions within a Flemish province were invited to participate. Seven of the eight institutions decided to participate in the study (participation rate = 87.5%), translating to a total number of 975 pre-service teachers filling out the online research instruments.

comment 7

In what cases were respondents deleted from the data: there were quite many (attrition rates of 33% and 15%). On p. 13 the authors write that means were computed on scales with a max of 25% missing values. This suggests that respondents with missing values are still in the database. So how many missing values per respondent were allowed?

Response 7

A consistent data maximalization approach was used throughout the analyses in order to maximize power for the analyses. Hence, the reviewer is correct in noting that respondents were not a priori deleted from the dataset for failing to provide answers to the final items in the survey. This was done as simulation studies have shown that when there are numerous missing values (as is the case for our data-sets), such listwise deletion leads to a biased sample and, hence, biased parameters and standard errors (see Enders, 2001, for an illustration). Rather, when data are not missing completely at random, the use of all available information with a maximum likelihood estimator has been shown to be the preferred option above list-wise deletion approaches or certain imputation techniques when performing factor analyses. Consequently, in many software packages, including MPlus which has been used in the current study, estimations that use all cases have been integrated as the default option. Hence, in the current study we have followed this advice by employing all available data for the factor analyses by not using a list-wise deletion approach to missing data and employing a maximum likelihood estimator (Newsom, 2023).

When it comes specifically to scale-construction, scales were only computed when the number of missings on items was 25% or less. This choice was made from a data maximalization perspective and in order not to penalize respondents who, for instance, accidentally skipped an item when filling out the survey. At the same time, in order to guarantee that scale-scores provided a reflection of scores on underlying items and would not be calculated for people who haphazardly ticked some boxes throughout the survey, we made sure that when respondents had more than 25% missing answers on item-scales, they received a missing for the overall scale-score. Note that these scale-scores were computed specifically for the comparison of the scale-scores as a final analysis (discussed as a final step in the Plan of Analysis and reported on pg 18 in the manuscript). Hence, these scale-scores were computed on the data from step two, being the data from in-service teachers which showed much lower attrition rates than the data from the pre-service teachers (which were only employed for the exploratory factor analyses in step 1). This can be seen in Table 4 “descriptive statistics of scale-scores” which shows the N of each scale employed in the ANOVA-analysis and which consists of 524 respondents for the efficacy-scales and 520 respondents for the beliefs-scale, and hence is quite comparable in number of respondents. In order to make these choices more clear to the reader, we have included some extra explanatory sentences to the Plan of analysis, as can be seen on page 13 in the manuscript and in the excerpt below.

To validate the factor structure underlying the DISCO-instrument, we combined exploratory (EFA) and confirmatory factor analyses (CFA) in a stepwise approach. In order to maximize power for the analyses and following the full information maximum likelihood on missing data theory (Newsom, 2023), all available data was used for the factor analyses. Hence, respondents were not a priori list-wise deleted from the data-set for having missings on certain survey-items. 

…

Finally, based on these results, the scale scores for in-service teachers were obtained by computing a mean for each respondent with a maximum of 25% missing values on the scale-items.

References for response to reviewers:

Ainscow, M., & Miles, S. (2008). Making Education for All inclusive: where next? Prospects, 38(1), 15-34. 

Billiet, J., & Waege, H. (2003). Een samenleving onderzocht: Methoden van sociaal-wetenschappelijk onderzoek. De Boeck Hoger. 

Burns, T., & Shadoina-Gersing, V. (2010). Educating teachers for diversity: meeting the challenge. Paris: OECD. 

Dyson, A., Howes, A., & Roberts, B. (2002). A systematic review of the effectiveness of school-level action for promoting participation by all students (EPPI-Centre Review). Research Evidence in Education Library. Issue 1.:. 

Enders, C.K. (2001). A primer on maximum likelihood algorithms available for use with missing data. Structural Equation Modeling, 8, 128- 141.

Gheyssens, E., Coubergs, C., Griful-Freixenet, J., Engels, N., & Struyven, K. (2020). Differentiated instruction: the diversity of teachers’ philosophy and praxis to adapt teaching to students’ interests, readiness and learning profiles. International Journal of Inclusive Education, 1-18. https://doi.org/10.1080/13603116.2020.1812739

Groenez, S., Vantieghem, W., Lamberts, M., & Van Avermaet, P. (2018). Diversiteitsbarometer Onderwijs Vlaamse Gemeenschap. In E. Keytsman (Ed.), Diversiteitsbarometer onderwijs (pp. 47–193). Unia. 

Messiou, K. (2017). Research in the field of inclusive education: time for a rethink? International Journal of Inclusive Education, 21(2), 146-159. 

Opertti, R., Walker, Z., & Zhang, Y. (2014). Inclusive education: From targeting groups and schools to achieving quality education as the core of EFA. In L. Florian (Ed.), The SAGE Handbook of Special Education (pp. 149-169). Sage. 

Pohan, C. A., & Aguilar, T. E. (2001). Measuring educators’ beliefs about diversity in personal and professional contexts. American educational research journal, 38(1), 159-182. 

Ponet, B., Tack, H., Vantieghem, W., Van Avermaet, P., & Vanderlinde, R. (2021). Hoe lerarenopleiders omgaan met diversiteit: naar een conceptueel raamwerk. Tijdschrift voor Lerarenopleiders, 42(4), 45-60. 

Van Avermaet, P., Van Houtte, M., & Van den Branden, K. (2011). Promoting equity and excellence in education. An overview. In K. Van den Branden, P. Van Avermaet, & M. Van Houtte (Eds.), Equity and excellence in education. Towards maximal learning opportunities for all students (pp. 1-20). Routledge. 

Van Mieghem, A., Verschueren, K., Petry, K., & Struyf, E. (2020). An analysis of research on inclusive education: a systematic search and meta review. International Journal of Inclusive Education, 24(6), 675-689.

---

## [Decision Letter · Decision Letter 1]

21 Aug 2023

Education for all in action: Measuring teachers’ competences for inclusive education

PONE-D-22-31586R1

Dear Dr. Vantieghem,

We’re pleased to inform you that your manuscript has been judged scientifically suitable for publication and will be formally accepted for publication once it meets all outstanding technical requirements.

Kind regards,

Pieter-Paul Verhaeghe

Academic Editor

PLOS ONE

Additional Editor Comments (optional):

Reviewers' comments:

Reviewer's Responses to Questions

**Comments to the Author**

1. If the authors have adequately addressed your comments raised in a previous round of review and you feel that this manuscript is now acceptable for publication, you may indicate that here to bypass the “Comments to the Author” section, enter your conflict of interest statement in the “Confidential to Editor” section, and submit your "Accept" recommendation.

Reviewer #1: All comments have been addressed

2. Is the manuscript technically sound, and do the data support the conclusions?

Reviewer #1: Yes

3. Has the statistical analysis been performed appropriately and rigorously? 

Reviewer #1: Yes

4. Have the authors made all data underlying the findings in their manuscript fully available?

Reviewer #1: Yes

5. Is the manuscript presented in an intelligible fashion and written in standard English?

Reviewer #1: Yes

6. Review Comments to the Author

Reviewer #1: Thank you to the authors for your thorough and well-reasoned response to our comments. I agree with you that this has made an already-strong paper even stronger. I am particularly pleased to see the additional explanations regarding the perspectives on inclusion, self-efficacy and competence, and to see the full list of questionnaire items. As before, the article is a good read, and I very much look forward to seeing it in print, and perhaps to using it with my students.

It is more of an editorial point, but I found Table 1 a bit difficult to follow, as it was not always clear which information aligned with which. This is partly because APA does not allow the use of horizontal lines. Perhaps shading the rows containing deleted items would help, and also making sure that the alignment in all cells is the same. I will leave it up to the editors to decide on this.

7. PLOS authors have the option to publish the peer review history of their article (what does this mean?). If published, this will include your full peer review and any attached files.

Reviewer #1: No

---

## [Editor Report · Acceptance letter]

27 Oct 2023

PONE-D-22-31586R1 

Education for all in action: Measuring teachers’ competences for inclusive education 

Dear Dr. Vantieghem:

I'm pleased to inform you that your manuscript has been deemed suitable for publication in PLOS ONE. Congratulations! Your manuscript is now with our production department. 

Kind regards, 

on behalf of

Professor Pieter-Paul Verhaeghe 

Academic Editor

PLOS ONE